# Engineering programmable CAR and antigen pairing via drug-gated light activation

Ziliang Huang [1,2,3], Praopim Limsakul[2,4], Yiqian Wu [2,5], Tianze Guo[1], Yuxuan Wang[1], Zhuohang Wu[1], Linshan Zhu[1], Molly E. Allen[2], Longwei Liu [1,2] & Yingxiao Wang [1,2] ✉

Chimeric antigen receptor (CAR) T-cell immunotherapy has achieved clinical success, but remains limited in solid tumors by antigen escape and tumor heterogeneity. Here, we engineer a high-affinity R-phycoerythrin (PE)-binding monobody to create a PE-programmable CAR toward user-defined antigens sequentially or simultaneously. To reduce off-tumor toxicity, we implement a drug-gated light-activation approach (DGLA) strategy to spatially confine CAR function. We further couple DGLA-controlled tumor antigen presentation with synNotch-mediated programmable antigen-targeting CAR (sPAT) to establish programmable CAR-antigen pairing (DGLA-sPAT). This system induces clinically validated antigens on tumor cells as local 'training centers' that recruit and activate sPAT CAR T cells, enabling elimination of entire tumor populations through broadly expressed tumor antigens using PE-conjugated antibodies. In vivo, DGLA-sPAT manifests local T cell activation and potent tumor suppression with minimal off-tumor toxicity. Thus DGLA-sPAT provides a modular and spatially controlled framework to overcome antigen escape and heterogeneity while improving safety in solid-tumor CAR-T therapy.

Chimeric antigen receptor (CAR) T cell-based therapies have achieved tremendous success in treating blood tumors[1–4]. While these therapies have shown promising outcomes for solid tumors in some cases, there are still challenges that must be overcome before their broad application[5–11]. One major obstacle is antigen escape, where tumor cells down-regulate the expression of antigens targeted by CAR T cells, resulting in tumor relapse[4,12,13]. In addition, tumor heterogeneity is challenging for current CAR T therapies, most of which are single-antigen targeted[4]. CAR T cells that can simultaneously target multiple antigens have been developed and have demonstrated successful outcomes[14–17]; however, broadening the spectrum of target antigens also increases the risk of off-tumor toxicity, especially when the therapeutic cells are delivered systemically[18–20]. Constitutively active CAR

T cells are difficult to control once infused into patients and hence not well versed in addressing these adversities[21–23].

Programmable antigen targeting (PAT) CAR, which binds to an adapter molecule that can be conjugated to various antibodies against different tumor antigens[8,24–26], has been developed to address the antigen escape and tumor heterogeneity[19,27]. The PAT CAR designs offer several potential advantages, including: (1) flexibility in antigen choice, which can be challenging to predict due to the heterogeneous complexity of physiology in different patients; (2) versatility in applications that allow the same CAR T cell product to be used for different antigen-targeting demands. However, immunogenicity and off-tumor toxicity still present potential safety concerns for the PAT CAR designs[28–30].

[1]Alfred E. Mann Department of Biomedical Engineering, University of Southern California, Los Angeles, CA, USA. [2]Shu Chien-Gene Lay Department of Bioengineering, Institute of Engineering in Medicine, University of California San Diego, La Jolla, CA, USA. [3]State Key Laboratory of Ultrasound in Medicine and Engineering, Chongqing Medical University, Chongqing 400016, China. [4]Division of Physical Science, Faculty of Science, Prince of Songkla University, Songkhla, Thailand. [5]National Biomedical Imaging Center, College of Future Technology, Peking University, Beijing 100871, China. ✉e-mail: ywang283@usc.edu

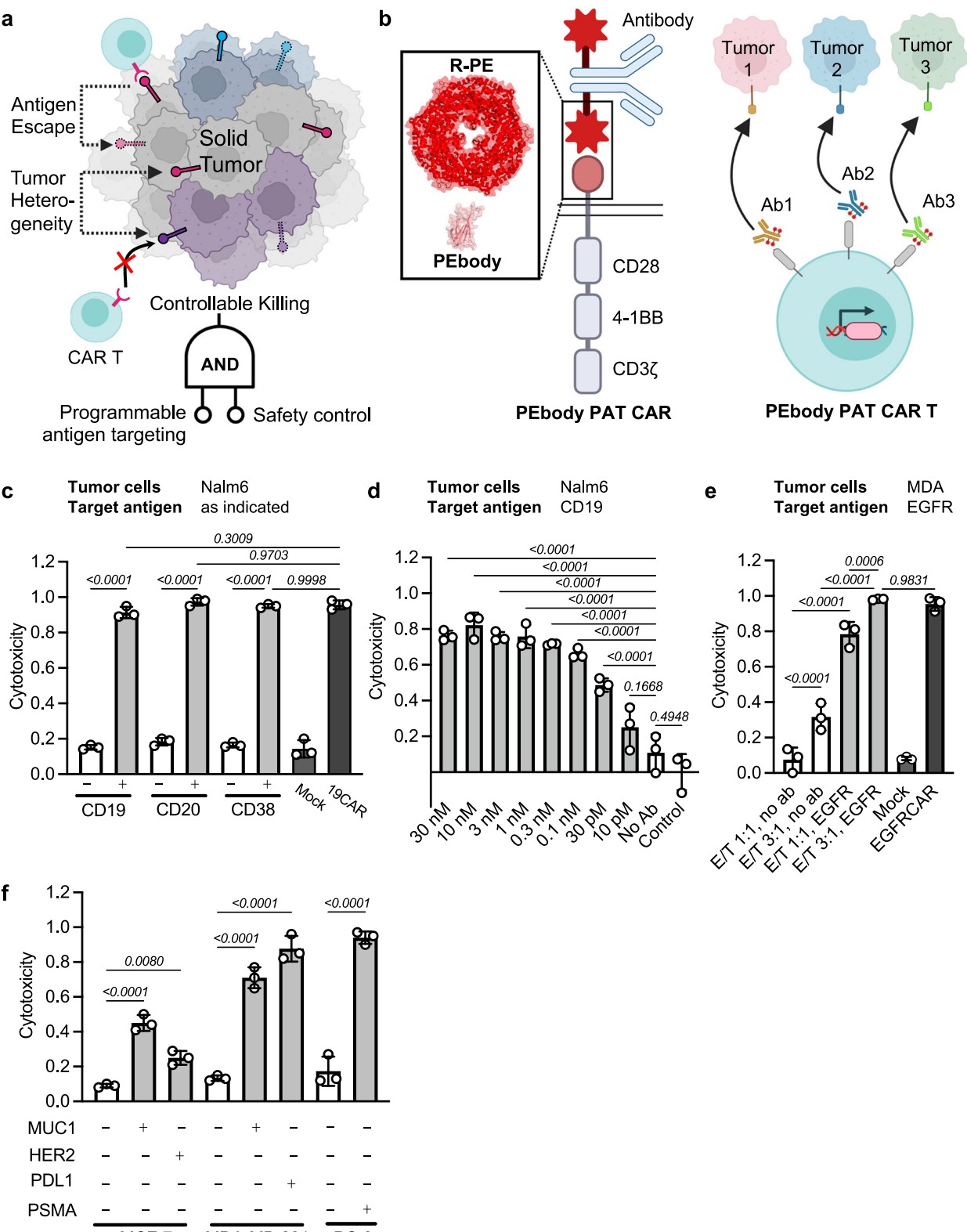

In this study, we aim to develop potent CAR T cells with programmable antigen targeting to address antigen escape and tumor heterogeneity and yet achieve safe CAR T therapies[31–34] (Fig. 1a). We developed a new PAT CAR based on PEbody, which is a monobody derived from human fibronectin type III via directed evolution by our group to be a specific binder of phycoerythrin (PE)[35]. This PEbody PAT CAR can be programmed to target multiple user-defined antigens via PE-conjugated antibodies sequentially or simultaneously. PE is

considered biocompatible and safe and has been widely used in food, cosmetics, pharmaceutical, and biomedical industries[36–38]. As such, compared with other PAT CARs that may cause immunogenicity[28–30], PAT CARs with the fully human-derived PEbody and its target PE can have better biocompatible immunogenicity[35,39,40].

To minimize the off-tumor side effects, especially when the PEbody PAT CAR T cells are empowered with broadened multi-antigen targeting capability, we have implemented the tamoxifen-gated photo-

**Fig. 1 | A PEbody-based PAT CAR. a** Current challenges of solid tumors include antigen escape and tumor heterogeneity, for which programmable antigen targeting (PAT) CAR design together with a safety control mechanism provides effective and safe solutions. Created in BioRender. Guo, T. (2026) https://BioRender.com/7kx4f7r. **b** Design of PEbody-based PAT CAR (PEbody PAT CAR). Left panel: the PEbody PAT CAR is composed of PEbody, which binds to PE-conjugated antibodies (enlarged schematic shown in box), CD28 (transmembrane and cytosolic domains), 4-1BB (cytosolic domain), and CD3ζ (cytosolic domain); right panel: the PEbody PAT CAR T cells can be programmed to target different antigens on various tumor cell types by their corresponding PE-conjugated antibodies. Created in BioRender. Guo, T. (2026) https://BioRender.com/5rg6elx. **c** Cytotoxicity of PEbody PAT CAR T cells against Nalm6 tumor cells when targeting different antigens (CD19, CD20, and CD38). E/T = 1:5. White bars, without antibody (as negative control); gray bars, with PE-conjugated antibody; mock, untransduced T cells; 19CAR, constitutive anti-CD19 CAR T cells. $n = 3$ biologically independent samples. Data are presented as mean values ±SD; one-way ANOVA with Sidak's

multiple comparisons test. **d** Cytotoxicity of PEbody PAT CAR T cells against Nalm6 cells at different antibody concentrations (PE-conjugated CD19 antibody). E/T = 1:5. Control, co-culture without CAR T cells. $n = 3$ biologically independent samples. Data are presented as mean values ±SD; one-way ANOVA with Sidak's multiple comparisons test. **e** Cytotoxicity of PEbody PAT CAR T cells against breast cancer cells MDA-MB-231 at varied E/T ratios as indicated. White bars, without antibody (as control); gray bars, with PE-conjugated EGFR antibody; mock, untransduced T cells; EGFRCAR, constitutive anti-EGFR CAR T cells. $n = 3$ biologically independent samples. Data are presented as mean values ±SD; one-way ANOVA with Sidak's multiple comparisons test. **f** Cytotoxicity of PEbody PAT CAR T cells against solid tumor cell lines MCF-7, MDA-MB-231, and PC-3. PE-conjugated antibody used for each test is as indicated. E/T = 1:1. White bars, without antibody (as control); gray bars, with PE-conjugated antibody, $n = 3$ biologically independent samples. Data are presented as mean values ±SD; one-way ANOVA with Sidak's multiple comparisons test. Cytotoxicity data were normalized to tumor-only control wells where no T cells were added. Source data are provided as a Source Data file.

activatable Cre (TamPA-Cre) approach to confine the PEbody PAT CAR T cell activity in the local tumor regions[41]. This drug-gated light-activation (DGLA) mechanism regulated by the FDA-approved drug tamoxifen can define a time window for intended light activation, thus minimizing Cre activity leakage and basal noise originating either from spontaneous pMag-nMag dimerization or unintended activation by ambient light sources[41,42]. This DGLA-controlled PAT CAR T approach (DGLA-PAT) should (1) allow targeting of user-defined tumor-associated antigens to overcome challenges in solid tumor therapies, such as antigen escape and tumor heterogeneity[6,10,12,43]; (2) provide the controllability of the therapeutic T cells' cytotoxicity in space and time thus minimizing the off-tumor toxicity[22,31,33]. We further extended the DGLA-PAT approach to combine DGLA-controllable tumor antigen presentation and synNotch-mediated PAT CAR (sPAT CAR) for programmable CAR-antigen pairing (DGLA-sPAT), which can induce clinically validated antigen presentation on the surface of cancer cells to serve as "training centers" for the recruitment and activation of sPAT CAR T cells. These trained or activated sPAT CAR T cells can then attack the entire cancer cell population at tumor sites, guided by PE-conjugated antibodies targeting broadly expressed tumor antigens. As these "training centers" can be precisely confined at local tumor regions by light, this DGLA-sPAT approach is not limited by targeting antigens that are exclusively expressed on tumor sites, which can be challenging to identify, while ensuring safe and efficient tumor clearance. Consistently, our in vivo tests in mice showed a local sPAT CAR T cell activation and effective tumor suppression with minimal off-tumor toxicity. As such, we have developed DGLA cells with programmable CAR and antigen pairing to target solid tumors with high flexibility, killing efficiency, and safety.

In summary, we establish a programmable CAR platform based on a PEbody that enables flexible multi-antigen targeting and combine it with a drug-gated light activation system to spatially restrict CAR activity. We further integrate inducible tumor priming with synNotch-regulated CAR expression to generate localized "training centers" that activate therapeutic T cells only within illuminated tumor regions. This strategy achieves effective tumor control in vivo while maintaining minimal off-tumor activity. Beyond this specific implementation, the framework provides a generalizable approach for decoupling antigen choice from CAR engineering and for enforcing spatial control over cell therapies. Such programmable and locally confined immunotherapies offer a path toward safer targeting of heterogeneous solid tumors and expand the design space for next-generation cellular therapies.

## Results
### A PEbody-based PAT CAR
We developed a PAT CAR using PEbody[35] and the third generation CAR design containing the human T cell co-stimulatory domains

CD28 and 4-1BB, and the intracellular signaling domain of CD3ζ[44] (referred to as PEbody PAT CAR). As such, PEbody PAT CAR can be programmed to target the PE tag on different antibodies recognizing their corresponding tumor-associated antigens (Fig. 1b, Supplementary Fig. 1a).

We firstly verified the programmable antigen-targeting capability of PEbody PAT CAR T cells using different PE-conjugated antibodies that target CD19, CD20, and CD38 on different tumor cell lines (Fig. 1c, Supplementary Fig. 1b). The orthogonality of PEbody PAT CAR was also confirmed (Supplementary Fig. 1c, d), demonstrating its ability to target various tumors and antigens with high specificity. Consistent with the PAT CAR design, the cytotoxicity of PEbody PAT CAR showed a dose-dependent pattern on antibody concentration (Fig. 1d). Increasing the effector-to-target (E/T) ratio also led to more effective killing of the target cells without significantly affecting the basal cytotoxicity (Fig. 1e, Supplementary Fig. 1e). To further demonstrate the broad applicability of PEbody PAT CAR, we tested its cytotoxicity against several solid tumor cell lines, including breast cancer cell line MCF-7 (targeting MUC1 or HER2), the aggressive triple-negative breast cancer cell line MDA-MB-231 (referred to as MDA, targeting MUC1 or PD-L1), as well as the prostate cancer cell line PC-3 (targeting PSMA), all demonstrating efficient killing performance with the corresponding target antibodies (Fig. 1f). Additionally, the PEbody PAT CAR T cells demonstrated robust upregulation of T cell activation markers CD25 and CD69 upon target engagement (Supplementary Fig. 1f), accompanied by cytokine secretion (Supplementary Fig. 1g). These results established that PEbody PAT CAR can be programmed by different antibodies to target corresponding tumor antigens with high specificity.

### Using PEbody PAT CAR T cells to mitigate antigen escape and heterogeneity of tumors
Antigen escape has become one of the major causes of relapse after CAR T therapies[45–47]. We tested the programmable feature of PEbody PAT CAR in an antigen-escape model of CD19, one of the prevailing causes of relapse after CD19CAR T treatment[45–47]. We reasoned that the same PEbody PAT CAR T cells can not only target CD19+ tumor cells, but also be redirected to target the CD19-negative cells in the event of antigen escape, by supplementing with other PE-conjugated antibodies such as those against CD20, another antigen on the target tumor cells (Fig. 2a). To mimic antigen escape, we generated CD19-negative Nalm6 cells via CRISPR-Cas9 mediated gene knockout (Fig. 2b, Supplementary Fig. 2a). We observed a decreased killing capability of PEbody PAT CAR T cells (targeting CD19) as the percentage of CD19-negative cells in the tumor population increased (antigen escape), which can be fully restored by switching to the PE-conjugated CD20 antibody while using the same CAR T cells (Fig. 2c).

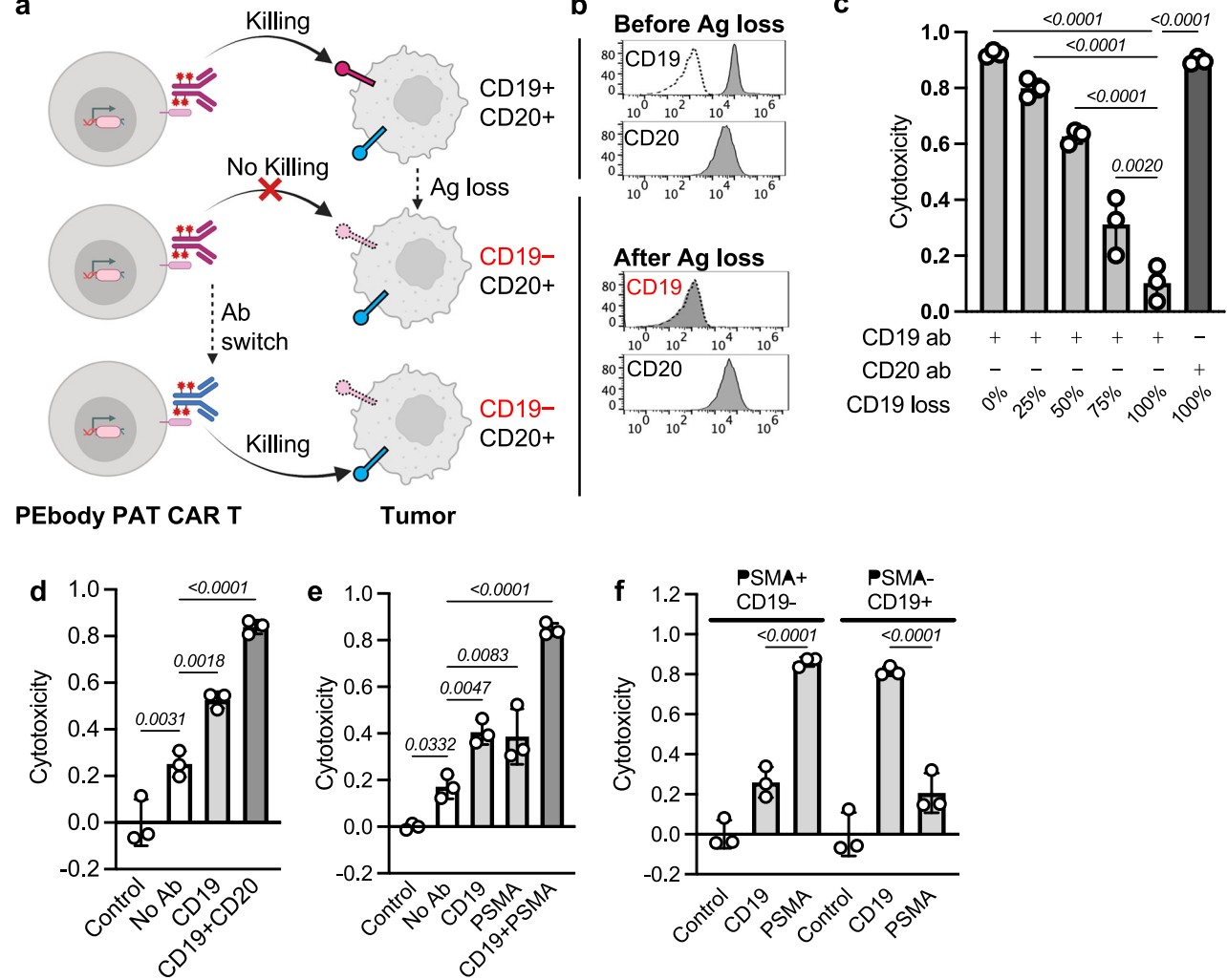

**Fig. 2 | PEbody PAT CAR T cells can be used to mitigate antigen escape and tumor heterogeneity. a** Schematics showing that PEbody PAT CAR T cells can still attack target tumor cells in the case of antigen escape using a CD19-based model. Top row, PEbody PAT CAR T cells can be programmed to target CD19 on the tumor cells (CD19+/CD20+); middle row, when antigen escape happens during therapy, the PEbody PAT CAR T cells previously programmed to target CD19 will lose the cytotoxicity against the target cell (CD19−/CD20+); bottom row, the same PEbody PAT CAR T cells can be re-programmed to target other tumor antigen(s) by switching the antibody (to CD20 antibody in this case), thus restoring the cytotoxicity against the "escaped" target cells (CD19−/CD20+). **b** Antigen expression profiles of Nalm6 tumor cells before and after antigen loss (generated by CRISPR-Cas9 knockout). Solid line, antibody staining; dotted line, CD19 isotype control staining. **c** Cytotoxicity loss of PEbody PAT CAR T cells against target cells (CD19+/CD20+) during CD19 antigen escape can be restored by re-programming the same PEbody PAT CAR T cells to target another antigen (CD20 in this case). E/T = 1:5, n = 3 biologically independent samples. Data are presented as mean values ±SD; one-way ANOVA with Sidak's multiple comparisons test. **d** PEbody PAT CAR T cells against

heterogeneous Nalm6 cells (CD19+:CD19− = 1:1). Control, co-culture without T cells; CD19, with PE-conjugated CD19 antibody; CD19 + CD20, with PE-conjugated CD19 and CD20 antibodies. E/T = 1:5, n = 3 biologically independent samples. Data are presented as mean values ±SD; one-way ANOVA with Sidak's multiple comparisons test. **e** PEbody PAT CAR T cells against heterogeneous PC-3 cells (PSMA+/CD19−: PSMA−/CD19+ = 1:1). Control, co-culture without T cells; CD19, with PE-conjugated CD19 antibody; PSMA, with PE-conjugated PSMA antibody; CD19 + PSMA, with both antibodies. E/T = 1:1, n = 3 biologically independent samples. Data are presented as mean values ±SD; one-way ANOVA with Sidak's multiple comparisons test. **f** Cytotoxicity of PEbody PAT CAR T cells against PC-3 cells with different antigen profiles (PSMA+/CD19− or PSMA−/CD19+). Left, PC-3 with PSMA+/CD19−. Right, PC-3 with PSMA−/CD19+. CD19, with PE-conjugated CD19 antibody; PSMA, with PE-conjugated PSMA antibody, E/T = 1:1, n = 3 biologically independent samples. Data are presented as mean values ±SD; one-way ANOVA with Sidak's multiple comparisons test. Cytotoxicity data were normalized to tumor-only control wells where no T cells were added. Source data are provided as a Source Data file. Created in BioRender. Guo, T. (2026) https://BioRender.com/noovp6s.

As PEbody PAT CAR can also be programmed to target multiple antigens simultaneously, we further tested its cytotoxicity against heterogeneous tumors. With a mixed population of CD19− and CD19+ Nalm6 cells (1:1), the PEbody PAT CAR T cells can kill ~50% of the total tumor cells when targeting CD19 only (using PE-conjugated CD19 antibody), while a significant killing (>80%) can be achieved with both CD19 and CD20 antibodies present in the co-culture (Fig. 2d). Consistently, using engineered PC-3 prostate cancer cell lines (either CD19+/PSMA− or CD19−/PSMA+, Supplementary Fig. 2b, c), the

PEbody PAT CAR T cells also demonstrated excellent cytotoxicity against the heterogeneous population (CD19+/PSMA−: CD19−/PSMA + = 1:1) only when both CD19 and PSMA antibodies were present (Fig. 2e), with each antibody contributing to the killing of its corresponding subpopulation (Fig. 2f).

**DGLA-PAT CAR T cells**
The PEbody PAT CAR framework enables targeting of multiple antigens, a property useful for combating antigen heterogeneity and

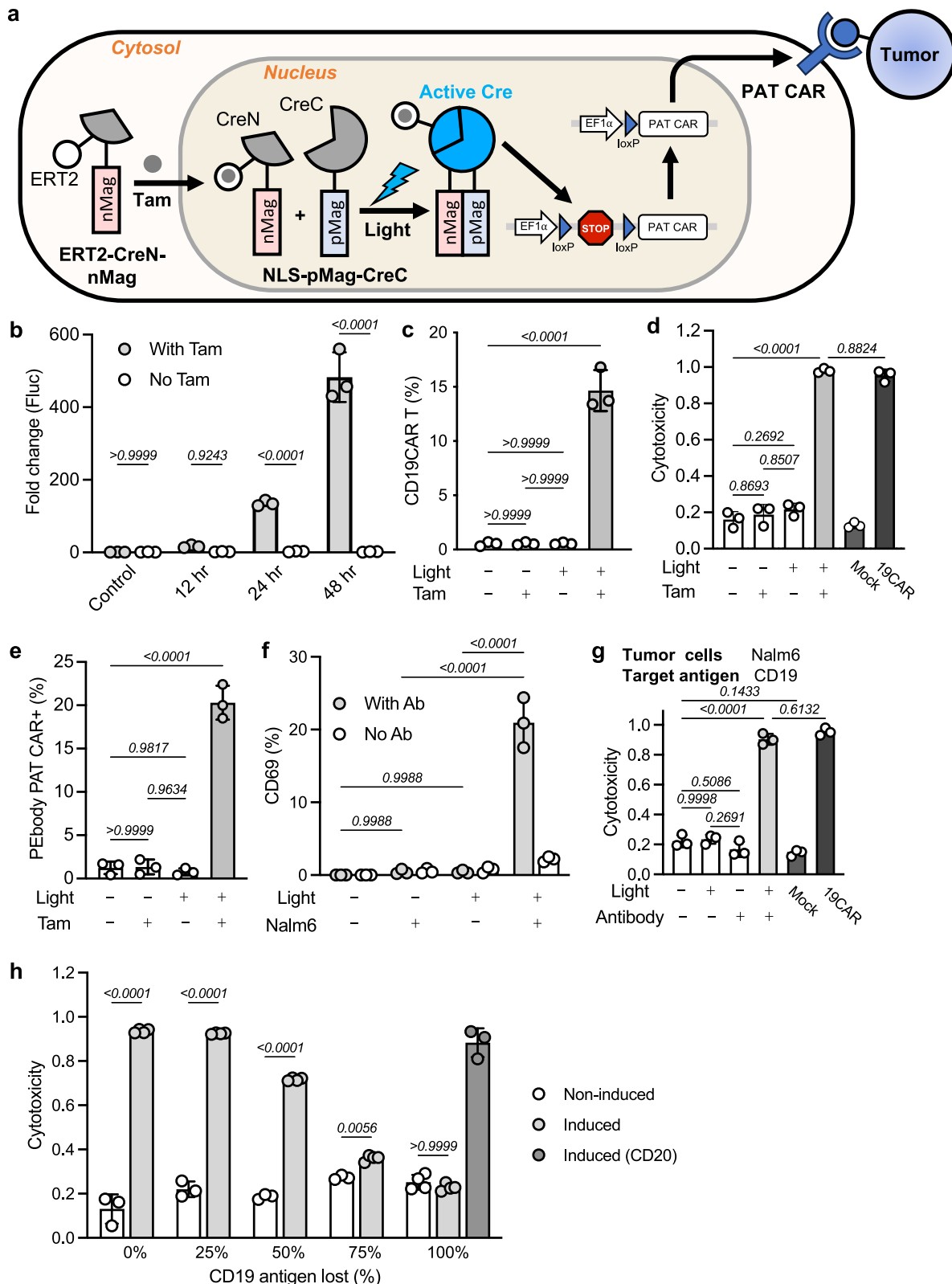

**a**

Cytosol

Nucleus

ERT2-CreN-nMag

NLS-pMag-CreC

Active Cre

PAT CAR

Tumor

We firstly demonstrated that bioluminescence reporter gene (Fluc) can be controlled by DGLA (Fig. 3b), and verified the inducible expression of CD19CAR in primary T cells (Fig. 3c), which can be translated into cytotoxicity against CD19+ target cells (Fig. 3d). Consistently, DGLA also allowed precise regulation of the PEbody PAT CAR expression in T cells by light (Fig. 3e), which can then be programmed

escape in solid tumors, but broad targetability can raise the risk of on-target/off-tumor effects[14–17]. To enhance safety while preserving flexibility, we integrated our drug (tamoxifen)-gated light-inducible (DGLA) gene expression system, TamPA-Cre[41], to develop DGLA-PAT CAR so that the action of PEbody PAT CAR T cells can be confined at tumor sites to reduce the off-tumor cytotoxicity (Fig. 3a).

**Fig. 3 | The cytotoxicity of DGLA PAT CAR T cells can be regulated by TamPA-Cre. a** Schematics of the Tamoxifen-gated Photo-Activatable Cre (TamPA-Cre or DGLA) system. The fusion protein ERT2-CreN-nMag consisting of human estrogen receptor mutant ERT2, N-terminal half of Cre (CreN), and nMag (from the light-inducible dimerization system Magnet) stays in the cytoplasm before priming by the tamoxifen metabolite 4-hydroxytamoxifen (4-OHT), which can facilitate the nuclear translocation of ERT2 and thus the whole fusion protein. Upon blue light stimulation, the nucleus-resided fusion protein NLS-pMag-CreC can dimerize with the translocated ERT2-CreN-nMag, thereby forming the intact and active Cre recombinase, which can trigger the gene switch to initiate expression of the gene of interest (e.g., PAT CAR). Created in BioRender. Guo, T. (2026) https://BioRender.com/rujhswd. **b** Test of DGLA gene expression using firefly luciferase (Fluc) gene as reporter in T cells. Tamoxifen metabolite 4-OHT was added in medium as 500 nM at 3 h before light stimulation. Blue light was applied as 5 s per min for 24 h; time points after start of light stimulation are as indicated. Control, without light stimulation; white, without tamoxifen (negative control); gray, with tamoxifen; $n = 3$ biologically independent samples. Data are presented as mean values ±SD; two-way ANOVA with Sidak's multiple comparisons test. **c** DGLA CD19CAR expression in T cells. $n = 3$ biologically independent samples. Data are presented as mean values ±SD; one-way ANOVA with Sidak's multiple comparisons test. **d** Cytotoxicity of DGLA CD19CAR T cells. T cells were co-cultured with Nalm6 cells (CD19+) after treatment for 24 h. Mock, untransduced T cells; 19CAR, constitutive anti-CD19 CAR T cells; E/T = 1:5. $n = 3$ biologically independent samples. Data are presented as mean values ±SD; one-way ANOVA with Sidak's multiple comparisons test. **e** DGLA

PEbody PAT CAR expression in T cells. Light (+/−), with or without light stimulation. Tam (+/−), with or without tamoxifen priming; $n = 3$ biological repeats. Data are presented as mean values ±SD; one-way ANOVA with Sidak's multiple comparisons test. **f** DGLA PEbody PAT CAR T can target tumor cells and lead to T cell activation. Light (+/−), with or without DGLA PEbody PAT CAR induction; Nalm6 (+/−), with or without target cells in co-culture; gray/white data points, with or without PE-conjugated CD19 antibody in co-culture; $n = 3$ biologically independent samples. Data are presented as mean values ±SD; two-way ANOVA with Sidak's multiple comparisons test. **g** Cytotoxicity of DGLA PEbody PAT CAR T cells against target cells (Nalm6, CD19+). Light/Tam (+/−), with or without DGLA (TamPA-Cre) induction. Antibody (+/−), with or without PE-conjugated antibody added in co-culture. T cells were co-cultured with Nalm6 cells after treatment for 24 h. Mock, untransduced T cells; 19CAR, constitutive anti-CD19 CAR T cells. E/T = 1:5, $n = 3$ biologically independent samples. Data are presented as mean values ±SD; one-way ANOVA with Sidak's multiple comparisons test. **h** DGLA-PAT CAR T cells can address antigen escape in a controllable manner. Different percentages of CD19 antigen loss were generated by mixing CD19+ and CD19− Nalm6 cells. DGLA-PAT CAR T cells with or without light induction were co-cultured with Nalm6 cells for 24 h. PE-conjugated CD19 antibody (light gray) or CD20 antibody (dark gray) was used for PEbody PAT CAR targeting. E/T = 1:5, $n = 3$ biologically independent samples in non-induced groups; $n = 4$ biologically independent samples in induced groups. Data are presented as mean values ±SD; two-way ANOVA with Sidak's multiple comparisons test. Cytotoxicity data were normalized to tumor-only control wells where no T cells were added. Source data are provided as a Source Data file.

to attack the target tumor cells, leading to controllable T cell activation (Fig. 3f) and cytotoxicity against target tumor cells (Fig. 3g). We further applied this DGLA-PAT CAR approach to address the antigen escape with the CD19 model introduced in Fig. 2a. Indeed, the "escaped" (CD19-negative) tumor cells can be contained by switching targeted antigen from the initial CD19 to CD20 (Fig. 3h), which can be precisely controlled by drug-gated light stimulation (induced vs. non-induced) to reduce off-tumor toxicity and achieve multi-antigen targeting, thereby addressing antigen escape and tumor heterogeneity in a single approach.

## DGLA tumor priming and PAT CAR killing in vitro
We further extended the potential of DGLA-PAT CAR for tumor priming and immunotherapy, using DGLA for local priming of tumor cells to guide the PAT CAR T cells for tumor killing. This delivery of DGLA and PAT CAR gene cassettes into separate cell types should reduce the genetic load for each cell type and thereby enhance the delivery efficacy (Fig. 4a). In this design, the tumor cells will be firstly vaccinated by a clinically validated non-pathogenic viral vector, adeno-associated virus (AAV)[48], to express DGLA and its inducible reporter for user-defined antigen presentation (e.g., clinically validated antigen, the truncated CD19 or tCD19 in this case). As tCD19 may likely be induced only in a subpopulation of the tumor cells due to limitation on efficiencies of gene delivery and DGLA activation, a direct targeting of the induced tCD19 could result in incomplete eradication of the entire tumor cell population. Instead, the subpopulation of tumor cells with induced tCD19 could be designed to serve as "training centers" for systemically delivered T cells engineered with anti-CD19 synthetic notch receptor (synNotch), and transactivate the expression of PEbody PAT CAR (synNotch-mediated PAT CAR, or sPAT CAR, Fig. 4b and Supplementary Fig. 3a). As such, the light-activated and primed subpopulation of tumor cells can "train" and activate the sPAT CAR T cells, which can be programmed to target antigen(s) of the whole cancer cell population at the local tumor site[6]. As the CAR expression will gradually decay when the sPAT CAR T cells leave the "vaccinated" and light-primed local tumor region, this will allow potent but less-ideal antigens to be used as targets for systemically introduced sPAT CAR T cells to ensure effective tumor killing with minimal off-tumor toxicity. We

named this specific strategy DGLA-sPAT CAR, with programmable CAR killing guidable by DGLA local priming of vaccinated tumors.

To test the DGLA-sPAT CAR system, HEK 293T cells were engineered to universally express PD-L1 to mimic the tumor antigen, as well as DGLA and its reporter, which can be induced to express tCD19 (Supplementary Fig. 3b). The sPAT CAR T cells were then introduced for co-culture, followed by the addition of PE-conjugated PD-L1 antibody to target the PD-L1+ cells (Supplementary Fig. 3c, upper panel). We observed over 90% tumor killing within 24 h in the group where target cells had been induced to express tCD19 (Supplementary Fig. 3c, lower panel), whereas the killing activity was minimized in the absence of antibody or DGLA induction. Similar killing effect by the sPAT CAR T was also observed when tCD19 was induced in the breast cancer cell line MDA-MB-231 expressing the endogenous PD-L1 (Fig. 4c).

We further tested the specificity of the DGLA-sPAT CAR approach using the engineered PC-3 cell lines (CD19+ or PSMA + PC-3, Supplementary Fig. 3d, e). While sPAT CAR T cells can efficiently kill CD19 + PC-3 cells (with antibody targeting CD19), they did not have significant cytotoxicity against CD19−/PSMA + PC-3 cells even in the presence of the PE-conjugated PSMA antibody, as PEbody PAT CAR cannot be induced. In the case of partial CD19 induction in a heterogeneous tumor population, as mimicked here by the mixture of CD19+/PSMA− and CD19−/PSMA + PC-3 cells (Supplementary Fig. 3f), the sPAT CAR T cells can still be sufficiently activated to target the heterogeneous population, with antibodies guiding the target antigens.

Due to the "training center" effect, we reasoned that the percentage and level of induced tCD19 in primed tumor cells and hence the light stimulation time can be less stringent while maintaining efficient killing. To test this, different lengths of light stimulation were applied to the tumor cells[41], and the expression of tCD19 was measured at the 24, 48, and 72 h time points after onset of light stimulation (Fig. 4d). We observed that the tCD19 expression was mainly affected by the illumination time duration, with moderate and gradual increase after illumination stopped (i.e., 48 and 72 h time points). Specifically, with 24 h illumination, over 85% of the tumor cells can be induced to express tCD19, in contrast to less than 1% tCD19 expressing cells in the group without light stimulation (Fig. 4e). Thus, antigen induction can

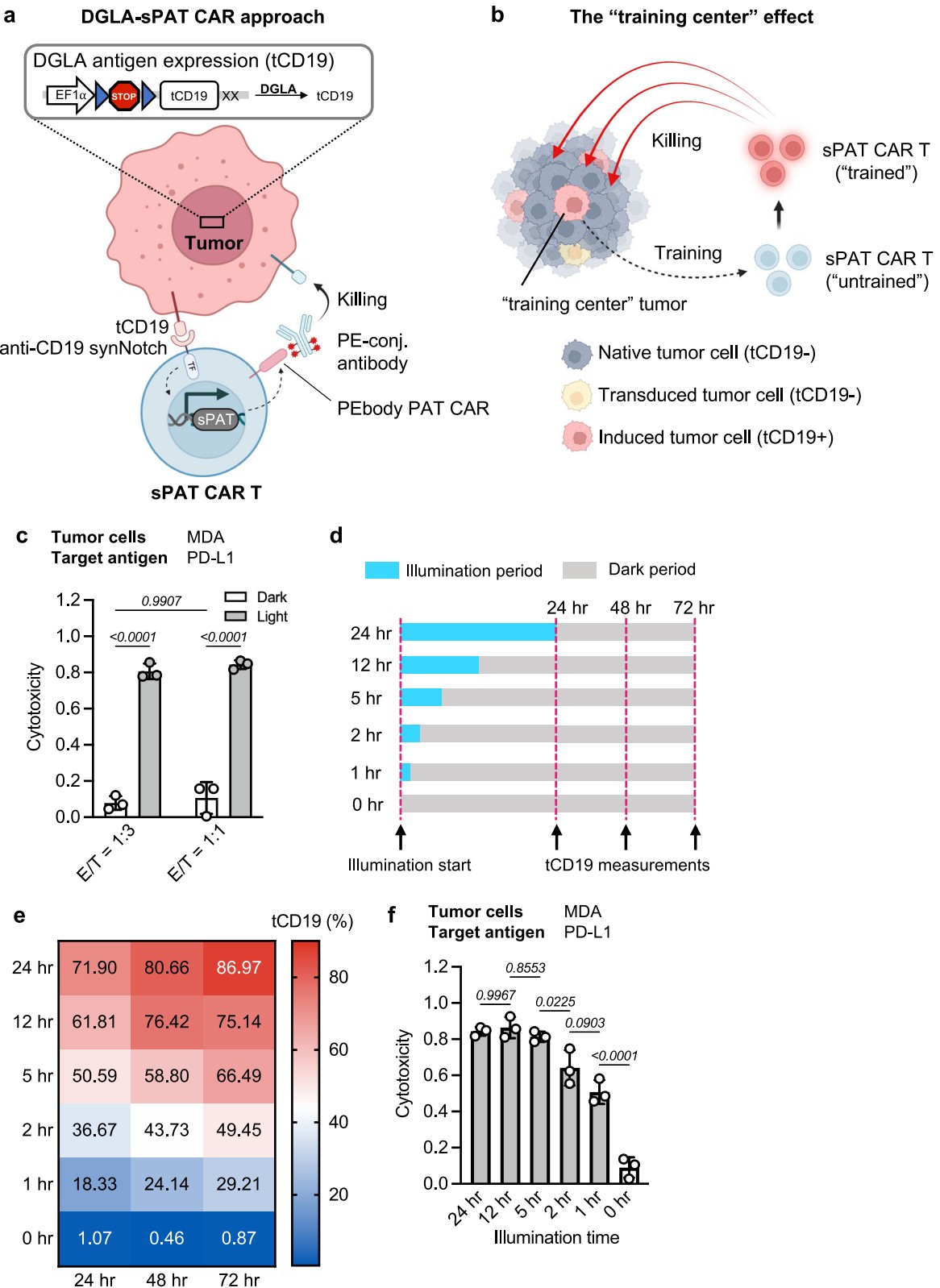

**a** DGLA-sPAT CAR approach

**b** The "training center" effect

**c** Tumor cells MDA / Target antigen PD-L1

**d**

**e**

**f** Tumor cells MDA / Target antigen PD-L1

be tightly controlled by light. Interestingly, we observed that over 50% of the tumor cells can be induced to express tCD19 with 5 h of illumination, which can be readily translated into effective killing through the sPAT CAR mechanism[6]. Indeed, we observed strong killing of target tumor cells under this shortened light stimulation, which is similar to that of the 24 h illumination group (Fig. 4f). Even with 1 h illumination, ~50% of killing can be obtained. These results suggest that DGLA-

sPAT CAR can achieve efficient tumor killing with flexible illumination conditions.

**DGLA-guided tumor priming in vivo**

We then tested whether DGLA can guide gene expression in mice. Nalm6 cells engineered with DGLA and the inducible firefly luciferase (Fluc) reporter were subcutaneously inoculated in the left and right

**Fig. 4 | DGLA-sPAT CAR and in vitro characterization. a** Schematics showing the mechanism of antigen presentation by DGLA and tumor killing by synNotch-mediated PEbody PAT CAR (DGLA-sPAT CAR). Local tumor cells can be "vaccinated" by compatible delivery vectors encoding DGLA and its inducible tCD19 reporter, the expression of which can be controllable in space and time by light. Once the CD19 synNotch-mediated PEbody PAT CAR (sPAT CAR) T cells engage with tCD19-expressing tumor cells, PEbody PAT CAR will be expressed on the T cells, which can be further programmed by user-defined antibodies to precisely target antigens expressed by the whole local tumor population. Created in BioRender. Guo, T. (2026) https://BioRender.com/p02b677. **b** Schematics of the "training center" effect in DGLA-sPAT CAR system. Each tumor cell that has been successfully transduced and induced to express tCD19 can activate multiple sPAT CAR T cells to express PEbody PAT CAR. Once activated ("trained"), each sPAT CAR T cell, in turn, can stay as active serial killer for tumor cells; combined with antibodies that target endogenous tumor associated antigens, the "trained" sPAT CAR T cells can now target the whole tumor population regardless of tCD19 expression. Thus, the tCD19-expressing tumor cells are acting as "training centers" for sPAT CAR T cells in our approach. Created in BioRender. Guo, T. (2026) https://

BioRender.com/cja9bdq. **c** Cytotoxicity of DGLA-sPAT CAR against MDA-MB-231 (MDA) cells. MDA cells were engineered with DGLA and inducible tCD19 reporter. Light/dark, with or without DGLA induction of MDA cells; E/T as indicated; $n = 3$ biologically independent samples. Data are presented as mean values ±SD; two-way ANOVA with Sidak's multiple comparisons test. **d** Scheme for DGLA mediated tCD19 induction and detection. Blue/gray, time periods with/without illumination. Length of illumination time under each condition is indicated on the left, pulsatile pattern (5 s light per min) was used during illumination period. Time points for tCD19 induction measurement: 24, 48, 72 h after illumination starts. **e** Induction of tCD19 after different lengths of illumination at the 24, 48, and 72 h timepoints after illumination starts. The percentage of tCD19-expressing cells is indicated in each condition (0–100%, color coded as from blue to red). **f** Cytotoxicity of DGLA-sPAT CAR against MDA cells, length of light stimulation as indicated. E/T = 1:3, $n = 3$ biologically independent samples. Data are presented as mean values ±SD; one-way ANOVA with Sidak's multiple comparisons test. Cytotoxicity data were normalized to tumor-only control wells where no T cells were added. Source data are provided as a Source Data file.

flanks of the same mice (Fig. 5a). For DGLA, we administered tamoxifen to the mice 1 day before light stimulation, as well as 5 h before light stimulation to ensure ERT2-regulated nuclear translocation of molecular regulators in tumor cells (Fig. 3a). We then applied pulsatile blue light (5 s per min) to one side of the mice for 12 h, while keeping the other side in the dark as control. Bioluminescence imaging was used here to monitor Fluc gene expression before and after light stimulation, with constitutively expressed Renilla luciferase (Rluc) to serve as normalization reference. Our results showed that blue light led to a ~35-fold increase of gene expression on day 3 when compared with the dark control (Fig. 5b). Moreover, the tamoxifen drug gating mechanism worked robustly in vivo, as no gene activation was observed with or without light stimulation in the absence of tamoxifen. Overall, our results demonstrated that DGLA can be used to achieve drug-gated light-controllable gene expression in vivo.

**Tumor homing and local activation of sPAT CAR T cells in vivo**
In parallel, we tested whether the systemically injected sPAT CAR T cells can home to and be activated locally at the tumor region in vivo that expresses the designed antigen (tCD19 in this case). To this end, one flank of the mice was inoculated with MDA-MB-231 cells expressing tCD19 ("tCD19+ MDA"), while the other flank inoculated with control MDA cells without tCD19 ("tCD19− MDA") (Fig. 5c). The expression of tCD19 did not affect the behavior of MDA tumor cells (Supplementary Fig. 4a). The CD19-targeting sPAT CAR T cells were then intravenously introduced into the mice. To visualize the homing and activation of sPAT CAR T cells, we also cloned the firefly luciferase gene into the synNotch reporter cassette downstream of the PEbody PAT CAR (Gal4UAS-Pmin-PEbody PAT CAR-P2A-Fluc, Fig. 5c enlarged panel from sPAT CAR T), so that activation of the sPAT CAR T cells can be visualized by bioluminescence in vivo[49]. Fluc bioluminescence was only observed at the side with "tCD19+ MDA" tumor cells following T cell injection, but not at the "tCD19− MDA" tumors (Fig. 5d, e, Supplementary Fig. 4b). The early rise in Fluc signal in Fig. 5e reflects synNotch activation of homed T cells rather than proliferation, as expansion-driven Fluc expression also requires subsequent PEbody CAR induction. These results suggest that the homing and local activation of sPAT CAR T cells only happened at the designated target tumor site (tCD19+), even though the control side mimicking the normal tissues (tCD19−) has similar antigen profiles except tCD19.

To assess whether the sPAT CAR T cells could suppress the target tumor ("tCD19+ MDA"), we also intravenously delivered PE-conjugated PD-L1 antibody 1 day after T cell injection, which can trigger the killing activity of the trained sPAT CAR T cells against MDA cells. Although MDA cells endogenously express PD-L1[50], tumor suppression was only

observed at the "tCD19+ MDA" tumor, with minimal effect observed at the "tCD19−" tumor (mimicking normal tissue expressing the same PD-L1 antigen as the tumor) (Fig. 5f, Supplementary Fig. 4c). The cytotoxicity of sPAT CAR T cells can hence be restricted to the local region of tumors expressing the clinically validated antigen, allowing the discrimination between tumors and normal tissues via the tCD19 antigen, thereby mitigating off-tumor toxicity after infusion.

**DGLA-sPAT CAR T immunotherapy in vivo**
We further examined the integrated DGLA-sPAT CAR T approach with MDA cells pre-engineered with the DGLA-inducible tCD19 (Fig. 6a), instead of constitutively expressing tCD19 as shown in Fig. 5c. After tumors were established, mice were primed with tamoxifen, and one side of the mice was subject to blue light stimulation as described earlier in Fig. 5b (referred to as "light side"), while the other side was kept in dark (referred to as "dark side"). Two days after light stimulation, sPAT CAR T cells were intravenously introduced, followed by injection of PE-conjugated PD-L1 antibody every 2 days to trigger tumor killing (Fig. 6b). Our results demonstrated that tumor growth on the light side was effectively suppressed by the sPAT CAR T cells, while the dark side remained unaffected (Fig. 6c, d). Similar results were observed in an additional study using the prostate cancer cell line PC-3 targeting the PSMA antigen (Supplementary Fig. 5), confirming the broad applicability of this approach. Furthermore, the DGLA-sPAT CAR T cells were detected in the peripheral blood and spleen after treatment, remaining non-activated (Supplementary Fig. 6a). In contrast, the tumor-infiltrating DGLA-sPAT CAR T cells were clearly activated with significant PAT CAR expressions (Supplementary Fig. 6b), consistent with the localized activation observed in tumors. More importantly, hematoxylin and eosin (H&E) staining of major organs alongside kidney and liver functional tests revealed no treatment-related lesions or toxicities (Supplementary Fig. 7), indicating the DGLA-sPAT CAR system's potential as a safe and effective treatment for solid tumors.

In addition to using pre-engineered tumor cells, we further assessed the system's applicability to tumors that are vaccinated by DGLA gene cassettes in situ, which better mimics clinical conditions. We generated tumors on both flanks of mice using wild type MDA-MB-231 cells, with one side serving as the target tumor and the other side mimicking a normal organ/tissue with a similar antigen expression profile (Fig. 6e). To achieve DGLA-guided tCD19 expression in the tumor cells, the adeno-associated viruses (AAVs) encoding the DGLA and the inducible tCD19 reporter were delivered intratumorally (Supplementary Fig. 8a, b). Two days later, mice were primed with tamoxifen and blue light was applied to the target tumor side to induce tCD19 antigen presentation (Fig. 6f). The sPAT CAR T cells were then

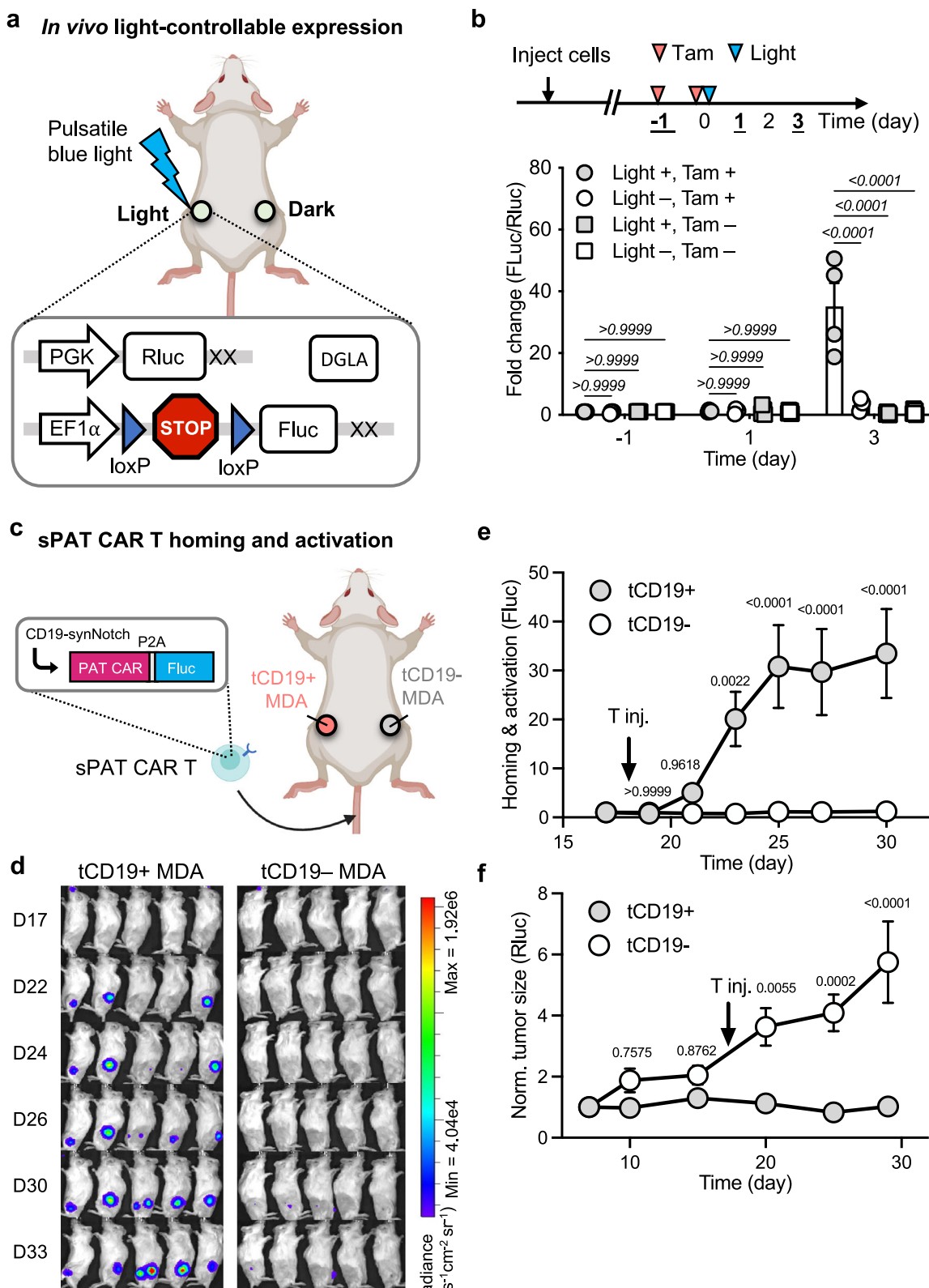

**a** *In vivo* light-controllable expression

**c** sPAT CAR T homing and activation

intravenously injected and subsequently "trained" and activated by the induced tCD19 in the tumor (Supplementary Fig. 8c) to express PEbody CAR. Subsequent injection of PE-conjugated PD-L1 antibody guided the activated sPAT CAR T cells to recognize and target PD-L1 on all MDA cells. Although this treatment was initiated at a later stage using more established tumors (day 15 after tumor inoculation), we still observed significant tumor growth slowdown and eventual reduction in size following the injection of sPAT CAR T cells and targeting antibody (Fig. 6g, h). In contrast, the control tumors mimicking normal tissues continued to grow exponentially, indicating minimal off-tumor toxicity of the DGLA-sPAT CAR approach. Another control experiment further confirmed that the suppression effect was specifically mediated by the designed sPAT CAR T cells (Supplementary Fig. 9).

**Fig. 5 | DGLA gene expression and sPAT CAR T cell homing and activation in vivo. a** Schematics of DGLA gene expression test in vivo. NOD/SCID/IL2rγ[null] (NSG) mice (8 weeks old, male, 4 per group) were subcutaneously injected with engineered Nalm6 cells expressing DGLA and the inducible firefly luciferase (Fluc) reporter. Constitutive *Renilla* luciferase (Rluc) was also expressed in Nalm6 cells as normalization reference. Blue light was applied on one side of the mice to induce gene expression while the other side was kept in dark. Created in BioRender. Guo, T. (2026) https://BioRender.com/ox0j5c8. **b** Upper panel: the timeline of DGLA gene expression test in mice. Tam, tamoxifen administration; Light, blue light stimulation. Lower panel: the fold change of gene expression (Fluc/Rluc) under different conditions. Light +/−, with or without light stimulation; Tam +/−, with or without tamoxifen administration. *n* = 4 biologically independent animals per group. Data are presented as mean values ±SEM; two-way ANOVA was used with Sidak's multiple comparisons. **c** Schematics of sPAT CAR T cell homing and

activation. NOD/SCID/IL2rγ[null] (NSG) mice (8 weeks old, female, 5 per group) were subcutaneously injected with "tCD19+ MDA" cells on one flank and "tCD19− MDA" cells on the other flank. Engineered T cells with anti-CD19 synNotch and inducible PEbody PAT CAR-P2A-Fluc cassette (enlarged panel) were introduced via tail vein. Created in BioRender. Guo, T. (2026) https://BioRender.com/ub375kt.
**d** Bioluminescence images of Fluc showing T cell homing and activation. The left ("tCD19+ MDA") and right ("tCD19− MDA") sides of the mice were shown respectively. **e** Quantification of T cell homing and activation based on Fluc bioluminescence. *n* = 5 biologically independent animals per group. Data are presented as mean values ±SEM; two-way ANOVA was used with Sidak's test. **f** Cytotoxicity of sPAT CAR T cells against MDA tumors. Tumor size was quantified based on constitutive Rluc bioluminescence. *n* = 5 biologically independent animals per group. Data are presented as mean values ±SEM; two-way ANOVA was used with Sidak's test. Source data are provided as a Source Data file.

To compare the safety and efficacy of the DGLA-sPAT approach with conventional CAR T therapy in vivo, particularly to evaluate mitigation of on-target off-tumor toxicity (OTOT), we generated a PSMA+ tumor model (Supplementary Fig. 10a), in which human PSMA was also expressed in the mouse liver via lentiviral transduction[20,51] (Supplementary Fig. 10b), followed by treatment with either DGLA-sPAT CAR T cells or conventional anti-human PSMA CAR T cells[52]. Successful hepatic expression of human PSMA was validated by in vivo and ex vivo bioluminescence imaging (Supplementary Fig. 10c, d). Although both conventional PSMA-CAR T and DGLA-sPAT CAR T effectively controlled tumor growth (Supplementary Fig. 10e–g), mice receiving conventional CAR T displayed apparent weight loss (Supplementary Fig. 10h), indicative of a heightened systemic toxicity burden[53]. Systemic cytokine profiling further revealed elevated cytokine levels in the conventional CAR T group (Supplementary Fig. 10i–k), consistent with an exaggerated inflammatory response[54]. Along with increased serum alanine aminotransferase (Supplementary Fig. 10l) and histological evidence of dense CAR-T infiltration and hepatocyte damage (Supplementary Fig. 10m), these findings indicate liver OTOT caused by conventional PSMA-CAR T treatment. In contrast, no significant OTOT-associated immune infiltration or tissue damage was observed in the examined organs from the DGLA-sPAT-treated mice, confirming its substantially improved safety profile. Collectively, these results demonstrate that DGLA-sPAT CAR T therapy achieves potent antitumor activity while spatially restricting CAR T cytotoxicity, enabling a generalizable and antigen-programmable platform for safe and effective treatment of solid tumors.

## Discussion

CAR T-based therapies have been successful in treating blood cancers[1,55], but solid tumor treatment remains challenging due to tumor heterogeneity, antigen escape, and toxic side effects[7,12,43,56]. Novel approaches have been developed to address these challenges, including drug-gated inducible CAR[57], combinatorial therapy[41], CAR design optimization[11], novel delivery systems[58], and gene editing technologies[5]. However, there is a lack of general approaches using CAR T cells for solid tumor treatment. In this study, we have developed a novel PEbody PAT CAR that allows for the same CAR design or even the same batch of CAR T cells to attack different tumor antigens/cell types as well as their combinations by programming and applying the corresponding antibodies. Furthermore, by integrating PAT CAR and DGLA gene expression, we developed DGLA-PAT CAR, which can confine the CAR T activation in the local tumor regions and reduce the off-tumor toxicities[21,41]. As such, DGLA-PAT CAR can mitigate antigen escape and tumor heterogeneity yet achieve a high level of safety and specificity. Further extension of DGLA-PAT by integrating DGLA for tumor priming and synNotch-mediated PAT CAR for killing (DGLA-sPAT) enabled programmable CAR-antigen pairing. This approach demonstrated strong efficacy and safety in

treating multiple tumor types, including triple negative breast cancer and prostate cancer.

It is a major challenge to identify an ideal antigen for solid tumors. As a result, standard CAR T cells can often lead to off-tumor toxicity, e.g., on-target off-tumor (OTOT) toxicity, which can be life-threatening[59,60]. We developed this DGLA-sPAT CAR approach to confine the CAR T functions at the local tumor sites, which will allow the flexibility to use strong but less ideal antigens as targets and achieve a low off-tumor toxicity. With the DGLA-sPAT CAR system, clinically validated antigens can be induced to express on cancer cell surfaces at local tumor regions, which can continuously recruit and train systemically administered sPAT CAR T cells in situ. The CAR levels will gradually decay when these activated sPAT CAR T migrate away from the tumor region, which can improve the safety of the treatment by reducing the adverse effects of off-tumor and OTOT toxicity. Although other inducible designs have shown potential in reducing off-tumor toxicity and exhaustion[5,6,22], concerns about efficacy and the requirement for repeated stimulations to achieve tumor suppression have been raised due to the transient induction[21,22,61]. Our study demonstrates the potential of DGLA, which allows for more sustained inducible expression to achieve long-lasting tumor killing with a single dose of stimulation[41,62], either by directly driving CAR expression on T cells or user-defined antigen expression on tumor cells.

Tumor heterogeneity and antigen escape are two challenges in treating solid tumors, in which tumor antigen expression can change at different time points and/or within different locations of the tumor region[63]. This has been reported to cause failures in multiple CAR T cell therapies[63,64]. Although CARs that can simultaneously target two or more antigens have been developed and reported to overcome tumor heterogeneity in some cases[27,65,66], substantial engineering and optimization efforts are needed to achieve positive outcomes. Our PEbody PAT CAR design allows for convenient targeting of multiple antigens simultaneously or sequentially without the need for further engineering[8,24,26], thus can provide flexibility in tailoring CAR T cell targeting specificity under different circumstances (Fig. 3d, e, Supplementary Fig. 1c). The PAT CAR design also allows for the switching of target antigens in case of antigen escape or relapse, by supplementing with different PE-conjugated antibodies at different stages of treatments, without the need for engineering a different CAR T product. Collectively, compared with traditional CAR-T approaches, DGLA-sPAT CAR offers two key advantages: light-gated spatial confinement of cytotoxicity to reduce on-target/off-tumor toxicity and programmable multi-antigen control to address heterogeneity and antigen escape.

The DGLA-sPAT CAR approach can benefit from the synNotch-inducible CAR expression, which can have intermittent rests when not engaged by the clinically validated antigens and alleviate the T cell exhaustion issue[5]. Indeed, constitutive CAR expression has been

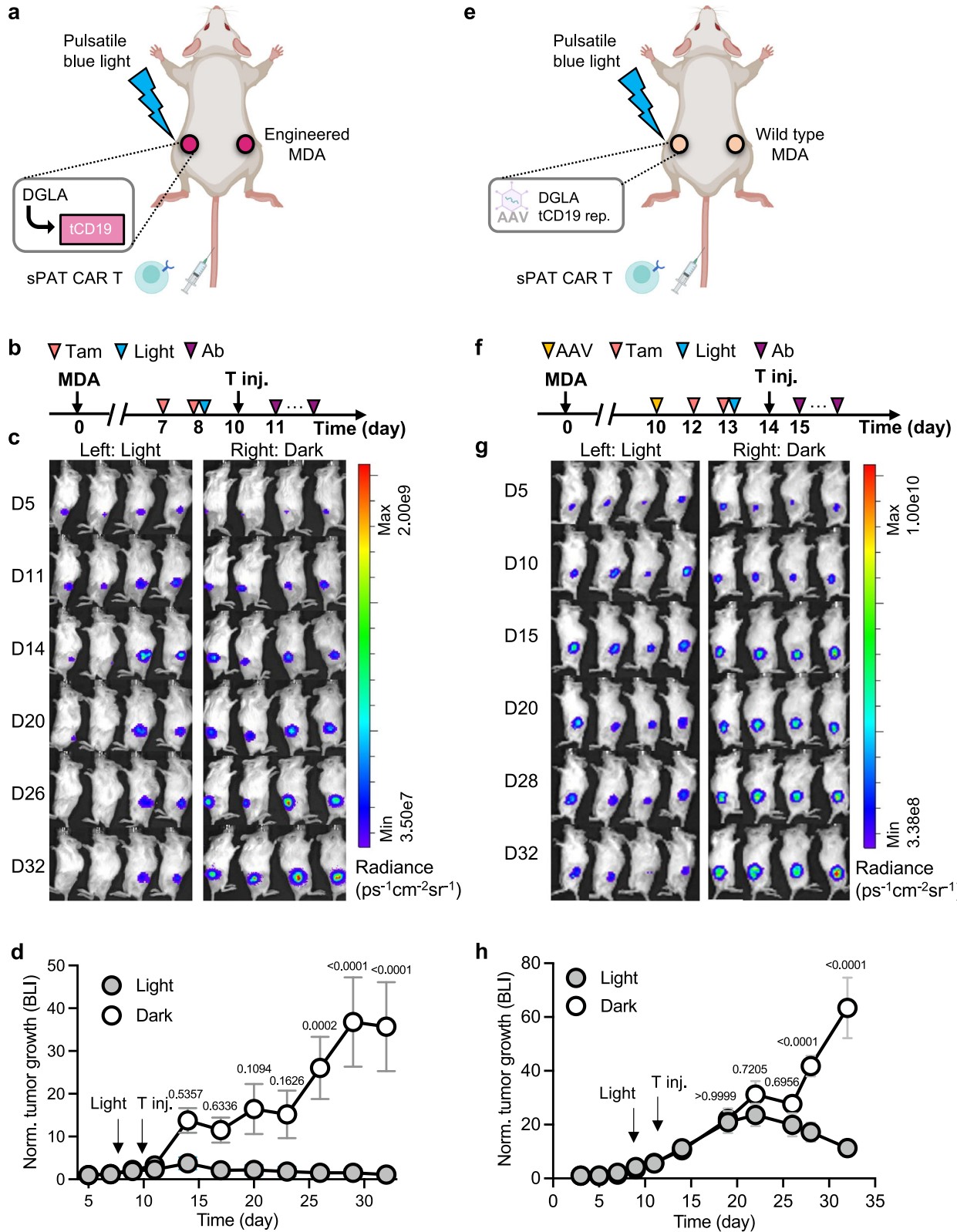

shown to cause tonic signaling and T cell exhaustion[67–69]. It has also been reported that transient rests or inhibition can restore the functionality of exhausted CAR-T cells[70]. As such, by integrating DGLA and PAT CAR, the DGLA-sPAT CAR system can provide a new tool for solid tumor treatment with enhanced persistence and suppressed tonic signaling and exhaustion, possibly by maintaining higher fractions of T cells potentially in the naïve or stem cell memory state[6].

Compared to the other PAT CARs such as the FITC CAR, PEbody was derived from human fibronectin type III domain[35] and thus the PEbody PAT CAR is humanized to have minimal risk of immunogenicity[71]. Although FITC has been a common fluorescent reagent for biological research, it has been reported to cause immune responses in vivo[28–30,72]. In contrast, PE is predominantly extracted from red algae and has been reported to be biocompatible and applied

**Fig. 6 | DGLA-sPAT CAR T immunotherapy in vivo. a** Schematics of DGLA-sPAT CAR T immunotherapy using engineered tumor cells. NOD/SCID/IL2rγ^null (NSG) mice (8 weeks old, female, 4 per group) were inoculated with MDA cells engineered with DGLA and inducible tCD19 reporter (enlarged panel). Blue light was applied on the target tumor to induce tCD19 expression after tamoxifen priming (5 s/min, 12 h). sPAT CAR T cells engineered with anti-CD19 synNotch and inducible PEbody PAT CAR reporter were delivered through tail vein followed by PE-conjugated antibodies. Created in BioRender. Guo, T. (2026) https://BioRender.com/eg8seu6. **b** Timeline of experiment as shown in (**a**). Mice were inoculated with engineered MDA cells on day 0. Tam, intraperitoneal tamoxifen administration. Light, blue light illumination on the tumor side. Ab, intravenous injection of PE-conjugated PD-L1 antibody. **c** Bioluminescence images of MDA tumor growth (Fluc). Left and right sides of the mice were shown respectively. **d** Quantification of tumor size of the left (with light stimulation) and right (without light stimulation) sides of the mice. $n = 4$ biologically independent animals per group. Data are presented as mean values ±SEM; two-way ANOVA with Sidak's test. **e** Schematics of DGLA-sPAT CAR T immunotherapy using tumor models vaccinated in situ. Wild type MDA cells were subcutaneously inoculated on left and right flanks of the same mice (NOD/SCID/IL2rγ^null (NSG) mice (8 weeks old, female, 4 per group). DGLA and the inducible tCD19 reporter were introduced into the tumor cells via AAV vectors. Blue light was applied on the target tumor to induce tCD19 expression (after tamoxifen priming). sPAT CAR T cells and antibodies were injected through tail vein. Created in BioRender. Guo, T. (2026) https://BioRender.com/va21rqk. **f** Timeline of experiments as shown in (**e**). Mice were inoculated with wild type MDA cells on day 0. AAV, intratumoral injection of AAV. Tam, intraperitoneal tamoxifen administration. Light, blue light illumination on the tumor side. Ab, intravenous injection of PE-conjugated PD-L1 antibody. **g** Bioluminescence images of MDA tumor growth (Fluc). Left and right sides of the mice were shown. $n = 4$ biologically independent animals per group. **h** Quantification of tumor size of the left (with light stimulation) and right (without light stimulation) sides of the mice. $n = 4$ biologically independent animals per group. Data are presented as mean values ±SEM; two-way ANOVA with Sidak's multiple comparison test. Source data are provided as a Source Data file.

to food industry[37]. In fact, PE has been demonstrated to have beneficial health effects, including antitumor activity tested in vivo and clinical trials[36,38]. Therefore, the PEbody PAT CAR can potentially reduce the immune-related risks associated with programmable and switchable CAR-based therapies.

In the DGLA-sPAT CAR T design, the expression of the tumor antigen that serves as the input for synNotch activation is not reliant on endogenous regulation, but rather induced synthetically via the DGLA system. Once triggered, this induction results in stable and constitutive expression of the synthetic antigen on the tumor cell surface. Furthermore, the DGLA gene cassette is stably integrated in the genome of tumor cells at a much higher copy number than that of the typical endogenous antigen genes. This makes antigen loss or down-regulation much more unlikely, thereby mitigating a major cause of resistance commonly seen in conventional CAR-T therapies.

The efficacy of PAT CAR approach is highly dependent on the expression level and conformation of target antigens on tumor cells. We observed that MCF-7 cells exhibited reduced sensitivity to PEbody CAR-mediated killing when targeted with anti-HER2 and anti-MUC1 antibodies compared to other cancer cell lines. The differential sensitivity of MCF-7 can be attributed to the unique characteristics of target antigen expression and presentation on this cell line. MCF-7 cells are well-established to express relatively low levels of HER-2 compared to other breast cancer cell lines[73,74], which directly impacts the availability of binding sites for anti-HER2 antibody and consequently reduces the efficiency of PEbody CAR T cell engagement. Similarly, while MCF-7 cells do express MUC1, the efficacy of anti-MUC1 antibody-mediated targeting is influenced by the complex interplay between antigen conformation and post-translational modifications. The glycosylation status of MUC1 has been shown to significantly modulate antibody binding affinity and specificity[75,76], with altered glycosylation patterns potentially masking or exposing different epitopes that affect antibody recognition. This dependency highlights the importance of careful target selection and potentially the need for combination approaches when targeting tumors with heterogeneous or suboptimal antigen presentation.

There are several optogenetic tools that have been or are being tested in clinical trials with potential against difficult-to-treat diseases[77–81]. One major potential risk in applying optogenetic tools for therapeutic purposes is immune responses towards the light-responsive proteins, which are typically derived from non-human organisms[21,33,34,82,83]. The DGLA system used in this study utilizes mutants of the VIVID photoreceptor from *Neurospora crassa*, which has been used in the food industry with a long history and no confirmed adverse immune response or toxicity in human has been reported[84], thus it may present lower immune-related risks compared to other light-inducible systems[21,33,34,82]. Additional options may also

become available in the future to overcome the immunogenicity challenge in the translational applications of the optogenetic systems. For example, to further reduce the immune-related risks, optogenetic systems can be used in immune-privileged organs (e.g., eyes, brains, testes, and placenta) or tolerogenic organs such as liver and spleen[85,86]. Additionally, novel approaches to circumvent pre-existing and induced immunity against protein-based therapeutics have been demonstrated effective with uncompromised performance, including structural modification of protein surface residues and using ortholog proteins from nonpathogenic organisms[87].

Although there is a limitation on penetration depth for blue-light based optogenetic systems, this could be overcome to a large extent in the future by using upconversion nanoparticles or implantable LEDs, which can convert near-infrared light or radio frequencies to shorter wavelengths (e.g., blue light) in vivo, and have been shown to successfully stimulate blue-light responsive systems in deep tissues[88,89]. Integrated with optical fibers, these light-activatable systems can also be extended to control T cell activity at gastrointestinal tracts for cancer therapies[90,91]. In summary, with unique advantages including precise activity control, long-lasting local effect, and convenient programmability, the DGLA PEbody-based CAR T approaches developed here should provide useful tools for solid tumor treatment particularly for locally advanced and unresectable solid tumors.

## Methods
### Ethics statement
Animal studies were conducted in compliance with all relevant ethical regulations and were approved by the Institutional Animal Care and Use Committee (IACUC) of University of Southern California (Protocol No. #21479). The maximal permitted tumor size/burden approved is 1.5 cm in diameter. Tumor growth was monitored 3 times per week, and this limit was not exceeded in all experiments. Animals were euthanized in accordance with the approved protocols if humane endpoints were reached. Sex was not considered in experiment design and analysis as outcomes were not expected to yield meaningful sex-specific differences, and the objectives of the study were not centered on sex-dependent effects.

### Cell culture
Human cell lines HEK 293T (ATCC CRL-3216), MCF-7 (ATCC HTB-22), MDA-MB-231 (ATCC HTB-26), and PC-3 (ATCC CRL-1435), all obtained from ATCC, were cultured in Dulbecco's Modified Eagle Medium (DMEM, Gibco, 11995073) supplemented with 100 U/mL penicillin-streptomycin (Gibco, 15140122), 2 mM L-glutamine (Gibco, 25030149), 1 mM sodium pyruvate (Gibco, 11360070), and 10% Fetal Bovine Serum (FBS, Gibco, 10438026). Human cell lines Jurkat (ATCC TIB-152) and Nalm6 (gift from Michel Sadelain lab, ATCC CRL-3273) were cultured in

RPMI 1640 (Gibco, 22400105), supplemented with 100 U/mL penicillin-streptomycin, 2 mM L-glutamine, 1 mM sodium pyruvate, and 10% FBS. Human peripheral blood mononuclear cells (PBMCs) were isolated from buffy coats obtained from the San Diego Blood Bank using lymphocyte separation medium (Corning, 25072CV) according to the manufacturer's instructions. The PBMCs were provided as fully de-identified, unidentifiable research-grade products. The study protocol and material usage were reviewed by the University of Southern California Institutional Review Board (IRB), which determined that the project does not meet the definition of human subjects research under 45 CFR 46.102(e)(1). Human T cells were subsequently isolated from purified PBMCs using a pan-T cell isolation kit (Miltenyi, 130-096-535). Human T cells were cultured in X-VIVO 15 medium (Lonza, 04-418Q) supplemented with 5% FBS, 10 µM 2-mercaptoethanol (Gibco, 31350010), and 100 IU/mL recombinant human IL-2 (PeproTech, 200-02), with Dynabeads Human T-Activator CD3/CD28 (Gibco, 11131D) following the manufacturer's protocol. All the cell cultures were maintained at 37 °C with 5% $CO_2$.

## Cloning

Primers for PCR amplification were synthesized by Integrated DNA Technologies (IDT Custom DNA Oligo service; sequences listed in Supplementary Table 1). PEbodyCAR construct (PEbody-CD28-4-1BB-CD3z): PEbody fragment was PCR-amplified from DNA template reported[35]; CAR fragment CD28-4-1BB-CD3z was amplified from a third generation anti-CD19 CAR (CD19 scFv-CD28-4-1BB-CD3z). TamPA-Cre (DGLA) constructs including Cre-loxP based CD19CAR reporter were PCR-amplified from constructs reported[41] (Addgene #192636 and #176311). The truncated CD19 (tCD19) reporter was constructed by replacing CD19CAR with tCD19 (amino acid 1-313) as amplified from Addgene #174610 with codon optimization for human cell line expression (Integrated DNA Technologies, CA). For lentivirus production, the lentiviral vector backbone from Addgene #79125 was used. For adeno-associated virus (AAV) production, the AAV2 vector backbone from Addgene #59462 was used. AAVs were produced by GT3 Core Facility of the Salk Institute. PCR amplifications were performed using Q5 high fidelity DNA polymerase (NEB, M0491S). DNA fragments were purified by agarose gel electrophoresis followed by column-based purification (Zymo Research, D4008). All the Constructs were generated by Gibson assembly (NEB, E2611L). Bacterial transformations were performed using NEB Stable Competent *E. coli* following the manufacturer's instructions (NEB, C3040H).

## Lentivirus production

HEK 293T cells were seeded 1 day before transfection in a 10-cm tissue culture dish and were transfected with 5 µg of psPAX2 (Addgene #12260), 5 µg of pMD2.G (Addgene #12259), and 10 µg of the corresponding transfer plasmid, using the ProFection Mammalian Transfection System (Promega, E1200). The medium was replaced with fresh medium at 6 h post-transfection. Lentivirus-containing supernatants were harvested at 48 h post-transfection, filtered through a 0.45 µm filter (Nalgene, 725-2545), and concentrated 100x in PBS with Lenti-X Concentrator (Takara, 631232). To determine the virus titer, HEK 293T cells were seeded at a density of 0.1 million cells per well in a 24-well plate and infected with serial dilutions of the concentrated virus. The titer was calculated as infectious units per milliliter (IFU/mL).

## Lentivirus infection

For infection of human cell lines, lentivirus was directly added to the culture medium with a multiplicity of infection (MOI) of 1–3, and the infection efficiency was checked at day 3 post infection. For the infection of primary human T cells, the naive CD3+ T cells were activated by CD3/CD28 Dynabeads (Gibco, 11141D) at 1:1 ratio for 48–72 h, and were infected in RetroNectin coated wells (Takara, T100B)

following a spinoculation protocol (1500 g, 1 h, 32 °C) with an MOI of 5–10. Infection efficiency was checked at day 3 post infection.

## Live-cell imaging

Primary human T cells transduced to express PEbody CAR-GGSGGT-eGFP (day 5–7 post-transduction) were washed and resuspended in phenol-red-free RPMI supplemented with 10 mM HEPES and 2% FBS. Live cells were imaged on a Nikon ECLIPSE Ti inverted microscope equipped with a CCD camera, using a 100× objective.

## Activation markers and cytokine measurements

A total of 0.1 million constitutive or inducible Jurkat T cells or primary T cells were co-cultured with 0.1 million Nalm6 cells with 3 nM PE-conjugated anti-human CD19 antibody (clone HIB90, mouse IgG1, κ, BioLegend 302208) in a 96-well plate for 24 h. Gating strategies were exemplified in Supplementary Fig. 11. Supernatants were collected for secreted IFN-γ, TNF, and IL-2 measurement via ELISA (Invitrogen 88-7316-22, 88-7346-22, 88-7025-22) following manufacturer's instructions. Cells were washed once with PBS, and resuspended in 100 uL PBS, and stained with APC-conjugated anti-human CD69 antibody (clone FN50, mouse IgG1, κ, BioLegend 310910) or APC-conjugated anti-human CD25 antibody (clone BC96, mouse IgG1, κ, BioLegend 302609) following the manufacturer's instructions (Supplementary Table 2). The stained cells were analyzed on a BD Accuri C6 flow cytometer.

## In vitro killing assays

Killing efficiencies of T cells were determined based on the firefly luciferase activity of the remaining target cells after co-culture using Bright-Glo Luciferase Assay System (Promega, E2610) following the manufacturer's instructions. PEbody PAT CAR T killing assay: 0.1 million T cells (engineered or non-engineered) were co-cultured with 0.1 million target cells expressing firefly luciferase with the corresponding PE-conjugated antibody in a 96-well plate for 24 h. To target different antigens, the following PE-conjugated antibodies were used: CD19 (clone HIB19, mouse IgG1, κ, BioLegend 302208), CD20 (clone 2H7, mouse IgG2b, κ, BioLegend 302306), CD38 (clone HB-7, mouse IgG1, κ, BioLegend 356604), MUC1 (clone 16A, mouse IgG1, λ, BioLegend 355604), HER2 (clone 24D2, mouse IgG1, κ, BioLegend 324406), PD-L1 (clone 29E.2A3, mouse IgG2b, κ, BioLegend 329706), PSMA (LNI-17, mouse IgG1, κ, BioLegend 342504). Data were normalized to tumor-only control wells where no T cells were added.

## CRISPR-Cas9 knockout of CD19 gene

Nalm6 cells were knockout by transfection with Cas9 protein-sgRNA (5′-CUAGGUCCGAAACAUUCCAC-3′, Invitrogen) RNP. Specifically, the SF Cell Line 4D-Nucleofector X Kit (Lonza) was used following the manufacturer's instructions. RNP was prepared by incubating mixture of 2 uL Cas9 protein (30 pmol/uL) and 3 uL sgRNA (20 pmol/uL) for 15 min at 37 °C. A total of 0.2 million of Nalm6 cells were transfected with the prepared RNP using program CV-104 in 20 uL Nucleocuvette Strips (Lonza, V4SC-2096). CD19 knockout on genetic level was verified by ICE analysis (Inference of Crispr Edits, Synthego). CD19 knockout on protein level was further verified by flow cytometry after immunostaining (clone HIB19, mouse IgG1, κ, BioLegend 302222) with isotype control (mouse IgG1, κ, BioLegend 400130), and CD19-negative clones were isolated by FACS sorting (Sony Biotechnology, SH800S) and amplified for later experiments.

## Antigen escape model

CD19-knockout Nalm-6 cell line was generated by CRISPR-Cas9, followed by verification of CD19 expression via flow cytometry. The 25%, 50%, 75%, and 100% CD19 loss cells were generated by mixing CD19-knockout Nalm-6 cells (0% CD19+) with wild-type CD19+ Nalm-6 cells

(100% CD19+) at the defined ratios to simulate varying levels of antigen escape.

## DGLA CAR T killing assay

T cells expressing DGLA and CAR reporter were treated with 500 nM 4-OHT (Sigma-Aldrich, H7904) for 3 h, followed by blue light illumination (5 mW/cm², 5 s per min, for 12 h). A total of 0.1 million T cells were co-cultured with 0.1 million target cells expressing firefly luciferase in a 96-well plate for 24 h. Luciferase activities were measured as described above.

## DGLA tCD19 based killing assay

Tumor cells expressing DGLA and tCD19 reporter were treated with 4-OHT for 3 h, followed by blue light illumination (5 mW/cm², 5 s per min, for 12 h). A total of 0.1 million anti-CD19 synNotch PEbody PAT CAR T cells were co-cultured with 0.1 million target cells (treated as described above or non-treated as control) expressing firefly luciferase in 96-well plate for 6 h, followed by addition of the corresponding PE-conjugated antibody and co-culture for another 24 h. Luciferase activities were measured as described above.

## Migration and invasion assays

MDA-MB-231 cell lines were starved for 12 h in 0.5% FBS DMEM, trypsinized and resuspended in 1% FBS DMEM. For migration, $5 \times 10^4$ cells in 0.25 mL were seeded into 24-well 8-µm PET inserts (Corning 353097); the lower wells contained 0.75 mL 10% FBS DMEM. For invasion, inserts were first coated on ice with 50 µL Matrigel Matrix (Corning 354248) diluted 1:3 in cold serum-free DMEM, polymerized 30–40 min at 37 °C, and hydrated 30 min in serum-free DMEM before seeding. All wells were incubated for 16 h at 37 °C, 5% CO₂. Percentages of migration/invasion were calculated using the counted cell number on the underside of inserts compared with the seeding cell number per insert.

## Serum cytokine, urea, and alanine transaminase (ALT) assays

Whole blood was collected from mice at indicated time points, allowed to clot for 30 min at 25 °C, and centrifuged at $2000 \times g$ for 15 min at 4 °C to obtain serum. Serum cytokine concentrations were quantified using ELISA kits following the manufacturer's instructions (Invitrogen IFN-γ: 88-7316-22; IL-2: 88-7025-22; TNF: 88-7346-22). Serum urea and alanine transaminase (ALT) levels were measured using the Urea Fluorometric Assay Kit (Cayman, #700620) and the Alanine Transaminase Colorimetric Activity Assay Kit (Cayman, #700260), respectively, according to the manufacturers' protocols.

## In vivo gene activation

Mice were housed in an SPF barrier facility under a 12-h light/12-h dark cycle at an ambient temperature of 20–26 °C and relative humidity of 40–60%. Mice were commercially purchased (no breeding) and experimental/control mice were housed in the same room under identical conditions but maintained in separate cages. Eight-week-old NOD/SCID/IL2rγ$^{null}$ (NSG, Jackson Laboratory, strain 005557) male mice were subcutaneously injected with 1 million Nalm6 cells engineered to express DGLA, its reporter loxP-ZsG-loxP-Fluc, and Rluc on the left and right flanks. For tamoxifen priming, tamoxifen (Sigma-Aldrich, 10540-29-1) was dissolved in corn oil (Sigma-Aldrich, C8267) at a concentration of 20 mg/ml by rotation at 37 °C until completely dissolved. A dose of 100 uL tamoxifen solution was administered through intraperitoneal injection at 24 h and 5 h before light stimulation. Blue light stimulation (460 nm, 5 W/m², 5 s/min) was applied for 12 h on the "light" side; and the "dark" side was protected from light. Bioluminescence imaging (BLI) was performed using an In vivo Imaging System Lumina LT Series III (PerkinElmer). Images were analyzed with Living Image software (PerkinElmer). The luciferase activity of Fluc and Rluc was measured using their respective substrates,

D-luciferin (GoldBio, LUCK-100) and coelenterazine (GoldBio, CZ10), following the manufacturers' protocols. To prevent interference between the two bioluminescence signals, the BLI of Fluc and Rluc in the same mouse was typically performed 4 h apart. Imaging settings were kept constant throughout the entire experiment. The maximum tumor size permitted is 1.5 cm in diameter, and all experiments were conducted in strict adherence to this limit. Tumor size was measured using digital calipers, and animals were monitored three times per week throughout the study. Criteria for early termination include significant body-weight loss (-15–20%), poor body condition or dehydration, inability to reach food or water, severe lethargy or unresponsiveness, persistent pain or distress, abnormal respiration or neurologic signs, large or ulcerated tumors exceeding approved size limits, serious infection or non-healing wounds, and any condition judged by veterinary staff to compromise animal welfare. Mice were euthanized by cervical dislocation by trained and experienced staff at experiment end points.

## In vivo T cell homing and activation

Eight-week-old NSG (Jackson Laboratory, strain 005557) female mice were subcutaneously injected with 1 million MDA-MB-231 cells expressing tCD19 (tCD19+) on one flank, and wild type (tCD19−) on the other flank, both expressing Rluc for tumor growth monitoring. For in vivo tCD19 induction experiments, 1 million of engineered DGLA-tCD19 MDA-MB-231 cells or DGLA-tCD19 PC-3 cells were injected on each flank of mice; all the tumor cells express Fluc for tumor growth monitoring. For in vivo gene delivery and tCD19 induction experiments, 1 million of wild type MDA-MB-231 cells were injected on each flank of mice; a total volume of 50 uL of purified AAV viral vector mixtures encoding DGLA and its reporter loxP-ZsGreen-loxP-tCD19 was injected intratumorally per tumor. All the tumor cells express Fluc for tumor growth monitoring. For DGLA activation, the tamoxifen priming and light stimulation were performed as described in the above section. When tumors are palpable, a total of 5 million human T cells transduced to express anti-CD19 synNotch and its reporter Gal4UAS-PEbodyCAR-P2A-Fluc (for synNotch activation monitoring and tumor killing) or Gal4UAS-PEbodyCAR (for tumor killing only) were injected through tail vein. BLI for Fluc and Rluc was performed as described above. For tumor killing, a dosage of 8 ug of anti-human PD-L1 antibody (for MDA tumor, clone 29E.2A3, mouse IgG2b, κ, BioLegend 329706) or anti-human PSMA antibody (for PC-3 tumor, clone LNI-17, mouse IgG1, κ, BioLegend 342504) was injected through tail vein every 2 days. The tumor size was monitored by Rluc or Fluc signals as indicated.

## Ex vivo flow cytometry and frozen-section histology/IHC

At endpoint, spleen, peripheral blood, and tumors were harvested. Single-cell suspensions were prepared as following procedures. Spleen: mechanical dissociation through a 70-µm strainer with ACK RBC lysis (Gibco A1049201, 2–5 min, RT); peripheral blood: EDTA-anticoagulated blood with ACK lysis (5 min, RT); tumor: mincing and digestion in DMEM containing collagenase D (Roche 11088858001, 1 mg/mL) and DNase I (Roche 04716728001, 0.1 mg/mL) for 30 min at 37 °C, quenched with 10% FBS and filtered (70 µm). Cells were washed in FACS buffer (PBS, 2% FBS, 2 mM EDTA). For staining, $1–2 \times 10^6$ cells were incubated 20 min at 4 °C with anti-human CD3-APC (clone OKT3; mouse IgG2a, κ; BioLegend 317317) and R-PE (Cayman 16637) to detect PEbody PAT CAR, washed, and resuspended in FACS buffer. Data were acquired on a BD Accuri C6 and analyzed in FlowJo v10. For histology, heart, liver, lung, spleen, kidney, and brain were embedded in OCT, snap-frozen on dry ice, stored at −80 °C, and cryosectioned at 10 µm onto Superfrost Plus slides. Sections were fixed in 4% paraformaldehyde (10 min, RT), rinsed in PBS, air-dried, and stained with hematoxylin and eosin (H&E). For immunohistochemistry of tCD19 on frozen tumor sections, tumors were harvested on day 8 after AAV

delivery and light stimulation, embedded in OCT, snap-frozen, and sectioned at 10 μm. Sections were fixed in 4% paraformaldehyde (10 min, RT), air-dried, quenched with 0.3% $H_2O_2$ in methanol (10 min), rinsed in PBS, and blocked in 5% BSA/PBS (30 min, RT). Slides were incubated with rat anti-CD19 (clone 6OMP31; Thermo Fisher 53-0194-82; 1:500 in blocking buffer) overnight at 4 °C, washed in PBS-T (0.05% Tween-20), incubated with HRP-conjugated goat anti-rat IgG (H + L) (Thermo Fisher 31470; 1:500) for 1 h at RT, developed with Pierce™ DAB Substrate (Thermo Fisher 34002) for ~2 min, rinsed, counterstained with hematoxylin, blued, dehydrated, cleared in xylene, and coated on coverslips. Imaging was performed on an EVOS M7000 (Thermo Fisher Scientific).

**In vivo comparison of DGLA-sPAT CAR-T and conventional CAR-T therapy**

Human PSMA was ectopically expressed in the livers of 8-week-old NSG male mice (Jackson Laboratory, strain 005557) by tail-vein injection of a lentiviral vector encoding PGK-nLuc-P2A-hPSMA ($1 \times 10^8$ TU/μL, 40 μL, diluted in 120 μL sterile DPBS). Hepatic nLuc expression was verified by luminescence imaging using the Nano-Glo fluorofurimazine in vivo substrate (Promega, N4100) according to the manufacturer's instructions. Seven days after hepatic transduction, mice were inoculated subcutaneously with $0.5 \times 10^6$ PSMA$^+$ PC-3 tumor cells expressing the DGLA-inducible tCD19 cassette and constitutive firefly luciferase. For the DGLA-sPAT group, tamoxifen (20 mg/mL in sterile corn oil; 75 mg/kg body weight) was administered intraperitoneally on days 11 and 12 after tumor implantation, followed by localized blue-light stimulation at the tumor site 4 h after tamoxifen on day 12 (460 nm, 5 mW/cm², 5 s/min, total 12 h). On day 14, all mice received a single intravenous infusion of $5 \times 10^6$ untransduced T cells (UTD), DGLA-sPAT CAR-T cells, or conventional PSMA CAR-T cells. Tumor burden was monitored by IVIS bioluminescence imaging following D-luciferin administration (GoldBio, LUCK-1G). Peripheral blood was collected at three time points, before infusion (pre-infusion), after DGLA induction (post-DGLA), and after CAR-T infusion (post-infusion), for quantification of systemic cytokines (human IFN-γ, IL-2, TNF; Invitrogen kits 88-7316-22, 88-7025-22, 88-7346-22). Serum ALT was measured post-infusion using the Cayman ALT Activity Assay (700260). Mice were euthanized when tumor diameter exceeded 1.5 cm or when ulceration or necrosis was observed. At study endpoints, major organs (heart, liver, lung, spleen, kidney) were harvested and processed for H&E staining to assess tissue pathology. For ex vivo liver bioluminescence imaging, mice were injected with fluorofurimazine substrate immediately before euthanasia, and freshly dissected livers were imaged using the IVIS system.

**Reporting summary**

Further information on research design is available in the Nature Portfolio Reporting Summary linked to this article.

## Data availability

Source data are provided with this paper. The cytotoxicity assays, T-cell activation marker expression, cytokine secretion measurements, bioluminescence imaging, tumor growth and in vivo therapeutic efficacy data, flow cytometry analyses, and histological/serum biochemistry data generated in this study are provided in the Supplementary Information/Source Data file. All data are included in the Supplementary Information or available from the authors, as are unique reagents used in this Article. The raw numbers for charts and graphs are available in the Source Data file whenever possible. Source data are provided with this paper.

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

## Acknowledgements

This work is supported by grants from NIH HL121365, CA262815, CA279813, EB029122, GM140929, and HD107206 (Y. Wang). This work was also supported by the GT3 Core Facility of the Salk Institute with funding from NIH-NCI CCSG: P30 014195, an NINDS R24 Core Grant and funding from NEI. The funding agencies had no role in the study design, data collection and analysis, decision to publish, or preparation of the manuscript.

## Author contributions

Z.H., P.L., and Y. Wang conceived and designed the experiments; Z.H., P.L., Y. Wu, T.G., Yuxuan W., Z.W., L.Z. and M.E.A. performed the experiments; Z.H., P.L., Y. Wu and L.L. analyzed the data; Z.H., P.L., L.L., and Y. Wang wrote the paper. All authors reviewed the manuscript and approved the final version.

## Competing interests

Y. Wang is scientific co-founder and consultant of Cell E&G Inc. and Acoustic Cell Therapy Inc. These financial interests do not affect the design, conduct or reporting of this research. The remaining authors declare no competing interests.
