## [Transparent Peer Review file · Nature Communications]

Engineering Programmable CAR and Antigen Pairing via Drug-gated Light Activation

Corresponding Author: Dr Yingxiao Wang

Version 0:

Reviewer comments:

Reviewer #1

(Remarks to the Author)

In this study, the authors developed a novel CAR that can be programmed with PE-conjugated antibodies to target multiple user-defined antigens, which allows for overcoming antigen escape and tumor heterogeneity. In this CAR-T system, by integrating PAT CAR and DGLA gene expression, DGLA-PAT CAR can confine the CAR T activation in the local tumor tissue and reduce off-target toxicity. This is a meaningful study. However, there are some experiments and controls that detract from the significance of this work. The following suggestions also should be taken into account.

1. The authors emphasize the local infiltration and activation of CAR-T and speculate that the expression of CAR-T diminishes after it migrates out of the tumor tissue. In animal models, it is recommended to supplement data related to the detection of CAR-T in peripheral blood, spleen, and draining lymph nodes after CAR-T treatment. In addition, it is recommended to add safety-related studies, such as liver and kidney function tests, as well as pathologic evaluation of important organs.
2. In Figure 2b, isotype as a control should be added to verify the knockdown effect of CD19 in Nalm-6 cells.
3. CD19 is a lineage marker for B cells, and the authors applied CD19+/PSMA-, as well as CD19-/PSMA+ prostate cancer PC-3 cells in their experiments, which required the detection of CD19 as well as PSMA expression in PC-3 cells.
4. In Figure 3, the authors constructed Tam-light induced PAT CAR-T and verified the ability to target kill target cells. However, the result does not seem to be sufficiently confirmed to attenuate the off-tumor toxic response to CAR-T therapy.
5. Figure S3b, Lack of statistical analysis results for left and right side tumors before treatment. From the results of Day 15, before treatment, the left side tCD19+ MDA tumors were smaller than the right side tCD19- MDA tumors, and the results of pre-treatment statistical analysis were required to ensure the reliability of the results. In addition, considering the possible effects of gene editing on MDA-MD-231 cells, the effects of expressing tCD19 on the proliferation, invasion and migration of MDA-MD-231 need to be verified.
6. In figure 3h, the results showed a difference in cytotoxicity between the non-induced and induced groups in CD19- Nalm-6 cells (100% loss of CD19 antigen). However, the non-induced group were normal activated T cells and the induced group were CAR-T cells that could target CD19. It is theoretically implausible that there is a difference in cytotoxicity between these two groups in CD19- cells.
7. The activation of CAR-T cells requires more detailed characterization, such as activation markers and cytokines.

Reviewer #2

(Remarks to the Author)

This manuscript from Ziliang Huang and colleagues described a strategy of T cell engineering for cancer therapy, which combines the applications of a PE-binding protein domain, the synNotch-regulated gene expression system, the tamoxifen-induced protein translocation system, and the light-responsive protein dimerization system. The authors aim to use the presented system to solve the current problems in T cell-mediated cancer therapy including the antigen loss, tumor heterogeneity, off-tumor toxicity, etc. They clearly showed that their DGLA-sPAT CAR system could work in experimental models based on CD19, PD-L1 as model antigens. Below are concerns that if addressed could expand the significance of the studies and offer clarification on certain points.

1. PEbody is generated by directed evolution. It's not clear whether PEbody can also bind cell/tissue in host, which means using it in a CAR might be risky for an on-target/off-PE effect. If this is the first case of using it in a CAR, there could be great value in investigating this attribute in the present study.

2. In the DGLA-PAT CAR design, tumor antigen is induced by light and Tamoxifen, PAT CAR expression is induced by synNotch recognition of induced tumor antigen, PAT CAR-T killing is induced by PE-antibodies. The concern is that due to multiple gatekeeping events, the eventual percentage of cancer-fighting CAR-T cells seems to be on the lower side of transferred CAR-T cells. According to Fig3c, e, f, the CAR induction is around 15-20%, activation is around 20%. Can the authors articulate or speculate as to why only a subset of these CARs are activated?

3. For the concept of "training center" tumor, the initial step is T cells expressing synNotch recognize either endogenous or DGLA-induced antigen, which means CAR-T cell activity is still dependent on one antigen, and antigen loss or downregulation will likely make such CAR-T therapy nonresponsive. Can the authors please address (defend or refute this assertion) this important issue either experimentally or in text.

4. In Fig 6, it seems that a clear interpretation of these data would require evaluating the effects of blue light treatment and AAV infection in one tumor side, control groups without sPAT CAR T cells.

5. Overall, the in vivo CAR T immunotherapy studies need to be strengthened. Although experiments described in Fig6 showed that DGLA-PAT CAR strategy could work to initiate tumor control based on inducible tCD19 expression and anti-PD-L1 treatment, there is no evidence that DGLA-PAT CAR is better than traditional CAR-T strategy in tumor control. And although the concept is novel, there is concern that given that the DGLA-PAT CAR strategy is more complicated in cell engineering, stricter in tumor killing initiation, possibly resulting in less tumor-fighting CAR T cells, a parallel comparison between DGLA-PAT CAR and traditional CAR is recommended. Can the authors please address (defend or refute this assertion) this important issue either experimentally or in text.

6. Only one tumor type, the MDA cell line, was used for in vivo tumor therapy study. To support the concept of "versatile" therapy strategy, multiple tumor types, antigens and antibodies need to be tested in more than one tumor model in vivo.

7. The description of the results is not clear. In the spirit of rigor and transparency, it is critical that experimental details be provided.

Minor concerns,

1. The surface expression of PEbody PAT CAR in T cells is not confirmed. This is important to know as this will impact signaling and ultimately efficacy.

2. In Fig1f, it seems that MCF-7 is not sensitive to killing by anti-HER2 and anti-MUC1. Please elaborate in the discussion.

3. Information of the antibodies used are not provided.

4. In Fig2c, details regarding the generation of 25%,50%,100% CD19 loss cell lines are not required.

5. In Fig3 and the verbiage in line 127-144 appears to be irrelevant to the topic of this manuscript. Also, although interesting, it is not clear how plausible it is for a CAR T therapy to engineer T cells with four transgenes for inducible CAR expression by Tam and light.

6. In Fig6, direct evidence for in situ AAV-injection and light-induced tCD19 expression in MDA tumor is not provided (although the concept is interesting). The PEbody CAR-T cell infiltration in tumor has not been thoroughly characterized nor has their function.

Reviewer #3

(Remarks to the Author)

This is a timely article reporting the development of a monobody based PE targeting CAR (PAT) that allows for T cell reactivity to be directed to tumor antigens via PE-conjugated antibodies. The authors demonstrate the ability of PAT T cells to kill tumor cells in a specific manner based on the presence of PE-conjugated antibodies. They further improve the CAR safety profile by engineering the PAT under a light/tamoxifen controlled Cre recombinase (DGLA-PAT CAR), which restricts CAR activity to local illuminated areas and requires the presence of tamoxifen. The authors also engineered a synNotch regulated PAT (sPAT), whereby the PAT expression is induced by priming of the SynNotch antigen. Most notably, the authors devised a CAR-antigen pairing system, whereby tumor antigen expression is controlled by the DGLA system and T cell reactivity is controlled by the sPAT modules (DGLA-sPAT). In this strategy, tumor cells can be vaccinated with an inducible antigen that can then "train" T cells to react specifically at the tumor site through the synNotch module. In murine flank models of human breast cancer, the authors show MDA cells engineered or vaccinated with the DGLA inducible CD19 cassette, express the target antigen only under illumination and with tamoxifen treatment. Administration of sPAT T cells and PE-conjugated PD-L1 antibody resulted in tumor regression only in tumors that were illuminated with light. These efforts are framed in the context of antigen escape, tumor heterogeneity, and safety concerns arising from clinical experience of standard CAR-T therapy, particularly for solid tumors. The impact and novelty of the manuscript is well suited for Nature Communications. The following revision experiments/edits would strengthen the manuscript:

1. In cytotoxicity assays (Figure 1 and Figure 3), the authors should include a conventional CAR control as a benchmark. It is unclear whether the inducible CAR formats reported in this study are of similar potency as standard CARs used in the field. The authors do a good job of motivating the clinical relevance of the study (i.e. addresses antigen heterogeneity, safety, and the biocompatibility of PE), but having a benchmark for potency is also needed to assess the clinical relevance. In addition to tumor killing, measuring the levels of cytokine secretion (i.e. IFN γ , IL-2) from tumor cocultures is appropriate for assessing

T cell functionality.

2. Similar to comment 1, Mock (untransduced) T cells are the appropriate negative controls in cytotoxicity assays that allow the reader to assess the leaky or non-specific activity of the system and should be included.

3. In Figure 3C, are the cells gated on transgene+? It is unclear if the relatively low induction of CD19 CAR-T (~15%) is because of inefficiencies of the inducible system or if it is due to low transduction efficiencies. The authors should clarify these details.

4. The cytotoxicity data presented in this study are normalized. The authors should indicate what the plots were normalized to.

5. In Figure 5E, it is unclear if the increase in fluc signal is driven by T cell expansion or synNotch activation since there is no constitutive and orthogonal luciferase that was coexpressed in the T cells. Did the authors administer PE conjugated antibody in this experiment? Adequate clarification or additional experiments should be provided to help interpret the source of the increasing fluc signal.

6. The authors should add a reference for the following study that demonstrated an inducible CAR regulated by tamoxifen activated recombinase.

Chakravarti, Deboki, et al. "Inducible gene switches with memory in human T cells for cellular immunotherapy." *ACS synthetic biology* 8.8 (2019): 1744-1754.

Version 2:

Reviewer comments:

Reviewer #1

(Remarks to the Author)

I thank the authors for their comprehensive responses to my previous comments and for the additional experiments provided, which have effectively addressed my concerns.

Specifically, the following:

1. Supplementary data on CAR-T cell detection in peripheral blood and spleen, as well as safety experiments including liver and kidney function tests and histopathological studies, strongly support the tumor-local activation characteristics and safety of this system.
2. The addition of isotype controls in the CD19 knockout experiment enhances the reliability of the results.
3. The validation of CD19 and PSMA expression in engineered PC-3 cells is now complete.
4. The explanation of the design change from DGLA-PAT to DGLA-sPAT and related in vivo experiments effectively address questions regarding the control of off-tumor toxicity.
5. Statistical analysis of pre-treatment tumor size and verification that tCD19 expression has no effect on cell behavior eliminate my concerns about potential experimental design.
6. The re-performed CD19-negative cytotoxicity experiment showed no difference between the two groups, consistent with theoretical expectations, making the conclusions more reliable.

In summary, the revised paper shows improvements in experimental completeness and clarity of argumentation. The method proposed in this study is innovative in improving the specificity and safety of CAR-T therapy.

I believe the authors have responded carefully to the reviewers' comments, and the paper meets the publication requirements; therefore, I recommend acceptance.

Reviewer #2

(Remarks to the Author)

All in all, the authors did a great job at addressing my (and it seems like the other reviewers') previous concerns. I commend them for their effort to address my concerns experimentally. This research is now much more robust. Overall, very nice work.

Reviewer #3

(Remarks to the Author)

Please add the ELISA data for conventional CAR in Figure S1G.

Otherwise, the revised manuscript has addressed my concerns and I recommend it for publication.

Version 3:

Reviewer comments:

Reviewer #3

(Remarks to the Author)

The authors have appropriately addressed my concerns in the revised manuscript. I recommend it for publication and congratulate the reviewers on the excellent work.

We thank the reviewers for their thorough evaluation of our manuscript and their constructive feedback. We have conducted additional experiments extensively and addressed all comments and suggestions, which have helped us improve the clarity and quality of our work. Below, we provide detailed responses to each reviewer's comments and describe the revisions made to address their concerns. We believe these changes have significantly strengthened the manuscript.

Reviewer #1 (Remarks to the Author)

In this study, the authors developed a novel CAR that can be programmed with PE-conjugated antibodies to target multiple user-defined antigens, which allows for overcoming antigen escape and tumor heterogeneity. In this CAR-T system, by integrating PAT CAR and DGLA gene expression, DGLA-PAT CAR can confine the CAR T activation in the local tumor tissue and reduce off-target toxicity. this is a meaningful study. However, there are some experiments and controls that detract from the significance of this work. The following suggestions also should be taken into account.

1. The authors emphasize the local infiltration and activation of CAR-T and speculate that the expression of CAR-T diminishes after it migrates out of the tumor tissue. In animal models, it is recommended to supplement data related to the detection of CAR-T in peripheral blood, spleen, and draining lymph nodes after CAR-T treatment. In addition, it is recommended to add safety-related studies, such as liver and kidney function tests, as well as pathologic evaluation of important organs.

Response: We appreciate this constructive suggestion and have added new data on CAR-T cell detection and safety-related studies. Briefly, we performed detection of DGLA-PAT CAR-T cells in peripheral blood and spleen after CAR-T treatment (**Fig. S6a**). The draining lymph nodes cannot be located due to their under development in NSG mice (Chappaz et al., J Immunol, 2010). The non-activated T cells, via anti-human CD3 antibody staining, are readily detectable in these locations, but none of them were activated as shown by the negative signals in PAT-CAR staining. Consistent with our design, the DGLA-PAT CAR-T cells were significantly activated in tumor region evidenced by the clear PAT CAR staining (**Fig. S6b**), confirming the intended tumor-specific activation.

Histopathology study, via H&E staining, of heart, lung, spleen, kidney, liver, and brain after CAR-T treatment (**Fig. S7a**), as well as liver and kidney functional tests were further performed (**Fig. S7b**). No treatment-related lesions or toxicities are observed. These results verified the safety of our approach. The Materials and Methods (line 538-558) and Results (line 249-254) sections have been updated accordingly.

2. In Figure 2b, Isotype as a control should be added to verify the knockdown effect of CD19 in Nalm-6 cells.

Response: Thank you for pointing this out. We have now included the isotype control staining result to verify the effect of CD19-knockout in the Nalm-6 CD19-KO cell line (**Fig. S2a**, line 112-113). The isotype control profile has also been included in the revised **Fig. 2B** (CD19 staining histogram), and noted in the figure legend.

3. CD19 is a lineage marker for B cells, and the authors applied CD19+/PSMA-, as well as CD19-/PSMA+ prostate cancer PC-3 cells in their experiments, which required the detection of CD19 as well as PSMA expression in PC-3 cells

Response: We have now validated the CD19 and PSMA antigen profiles of the engineered PC-3 cell lines and included the data in the revised manuscript (**Fig. S2b-c**, line 122-123).

4. In Figure 3, the authors constructed Tam-light induced PAT CAR-T and verified the ability to target kill target cells. However, the result does not seem to be sufficiently confirmed to attenuate the off-tumor toxic response to CAR-T therapy.

Response: We appreciate your insightful feedback regarding the evaluation of off-tumor toxicity attenuation in our study. In response, we would like to clarify two important points:

First, we have demonstrated in **Fig. 3g-h** that the cytotoxicity of DGLA PAT CAR-T cells can be effectively regulated through external stimuli such as light. These results indicate that the cytotoxicity of the engineered DGLA CAR-T cells can be precisely confined at the tumor site, supporting the potential for reducing off-tumor toxicity through on-demand activation such as light stimulation on tumor regions as shown in **Fig. 5c-f** and **Fig. 6**.

Second, due to the low efficiency of delivering all the three gene cassettes into primary T cells, it would be difficult to perform a direct *in vivo* validation with this initial design. Recognizing

these limitations, we subsequently transitioned to an alternative design that is more suitable for in vivo evaluation (**Fig. 4a-b**), as detailed in the later sections of our manuscript, which demonstrated attenuated off-tumor toxic response to CAR-T therapy (**Fig. 5c-f** and **Fig. 6**).

5. Figure S3b, Lack of statistical analysis results for left and right side tumors before treatment. From the results of Day15, before treatment, the left side tCD19+ MDA tumors were smaller than the right side tCD19- MDA tumors, and the results of pre-treatment statistical analysis were required to ensure the reliability of the results. In addition, considering the possible effects of gene editing on MDA-MD-231 cells, the effects of expressing tCD19 on the proliferation, invasion and migration of MDA-MD-231 need to be verified.

Response: We appreciate the reviewer's concern regarding potential baseline differences in tumor size prior to CAR-T-cell treatment. We have performed the statistical analysis on tumor growth comparing the left and right sides, each normalized to their respective Day 7 measurements, to evaluate any pre-treatment imbalance. As shown in **Fig. 5f** (associated with IVIS images of **Fig. S4c** in the revised manuscript), there was no statistically significant difference between the left side (tCD19+) and right side (tCD19-) MDA-MB-231 tumors prior to CAR-T-cell injection (noted as "ns" in **Fig. 5f**, $p > 0.05$). A significant divergence emerged only after CAR-T treatment, confirming that the therapeutic effect, rather than an initial size bias, accounts for the observed difference on tumor growth.

Regarding the potential effects of gene editing and tCD19 expression on MDA-MB-231 cell behavior, we respectfully note that tCD19 is a truncated, non-signaling surface marker that has been extensively validated as a clinically safe selection marker (Di Stasi et al., N Engl J Med, 2011; Terakura et al., Blood, 2012; Lee et al., J Immunother Cancer, 2023). It lacks intracellular signaling domains and has no known impact on cell proliferation, migration, or invasion. This inert nature has been demonstrated in multiple cell types, including T cells and hematopoietic stem cells, without affecting their biological behavior (Park et al., Sci Transl Med, 2020). We have also performed a comparison of proliferation, migration, and invasion between tCD19+ and tCD19- MDA-MB-231 lines, and observed no significant difference (**Fig. S4a**, line 217-218). Therefore, the use of tCD19 in our study is unlikely to alter the intrinsic properties of MDA-MB-231 cells.

6. In figure 3h, the results showed a difference in cytotoxicity between the non-induced and induced groups in CD19- Nalm-6 cells (100% loss of CD19 antigen). However, the non-induced group were normal activated T cells and the induced group were CAR-T cells that could target CD19. It is theoretically implausible that there is a difference in cytotoxicity between these two groups in CD19- cells.

Response: We thank you for this critical note. As correctly noted, CD19- Nalm-6 cells (with complete loss of CD19 expression) should not exhibit differential sensitivity to non-induced T cells versus induced PEbody CAR-T cells, as the antigen is absent. The small but statistically significant difference observed in the initial experiment is possibly attributable to experimental variation rather than true antigen-specific cytotoxicity. To clarify this point, we have re-conducted this assay with more biological repeats ($n = 4$). The updated results, now included in the revised **Fig. 3h**, show no significant difference between these two groups, consistent with the PEbody CAR design.

7. The activation of CAR-T cells requires more detailed characterization, such as activation markers and cytokines.

Response: We thank you for this important suggestion. To provide a more comprehensive characterization of CAR-T cell activation, we have performed additional experiments to characterize surface activation markers and cytokine release. Specifically, we analyzed the expression of CD69 and CD25 by flow cytometry (**Fig. S1f**), and measured the secretion of key cytokines including IFN- γ , TNF- α , and IL-2 via ELISA (**Fig. S1g**). These results further confirm the antigen-specific activation of CAR-T cells and are now included in the revised manuscript (line 101-103).

Reviewer #2 (Remarks to the Author)

This manuscript from Ziliang Huang and colleagues described a strategy of T cell engineering for cancer therapy, which combines the applications of a PE-binding protein domain, the synNotch-regulated gene expression system, the tamoxifen-induced protein translocation system, and the light-responsive protein dimerization system. The authors aim to use the presented system to solve the current problems in T cell-mediated cancer therapy including the antigen

loss, tumor heterogeneity, off-tumor toxicity, etc. They clearly showed that their DGLA-sPAT CAR system could work in experimental models based on CD19, PD-L1 as model antigens. Below are concerns that if addressed could expand the significance of the studies and offer clarification on certain points.

1. PEbody is generated by directed evolution. It's not clear whether PEbody can also bind cell/tissue in host, which means using it in a CAR might be risky for an on-target/off-PE effect. If this is the first case of using it in a CAR, there could great value in investigating this attribute in the present study.

Response: We thank you for highlighting this important point. In the current work, we have assessed the binding specificity of PEbody when used as the antigen binding motif in CAR against several human cell lines originated from different tissues including blood (Nalm-6, **Fig. 1c**), breast (MCF-7, MDA-MB-231, **Fig. 1f**), and prostate (PC-3, **Fig. 1f**). Our results showed that when PE-conjugated antibody is absent, there is no binding/killing against the target cells, indicating the binding of PEbody towards host cells/tissues is minimal. To further address the potential concern of “on-target/off-PE” effects in vivo, we performed histological studies of important organs, including heart, lung, spleen, kidney, liver, and brain (**Fig. S7a**), as well as liver and kidney functional assays (**Fig. S7b**), after PEbody PAT CAR T treatment. Our results revealed no signs of tissue damage or inflammation, supporting the conclusion that PEbody CAR-T cells do not exhibit detectable off-PE toxicity in host tissues. These new results have been included in the revised manuscript (line 252-254).

2. In the DGLA-PAT CAR design, tumor antigen is induced by light and Tamoxifen, PAT CAR expression is induced by synNotch recognition of induced tumor antigen, PAT CAR-T killing is induced by PE-antibodies. The concern is that due to multiple gatekeeping events, the eventual percentage of cancer-fighting CAR-T cells seems to be on the lower side of transferred CAR-T cells. According to Fig3c, e, f, the CAR induction is around 15-20%, activation is around 20%. Can the authors articulate or speculate as to why only a subset of these CARs are activated?

Response: We appreciate your thoughtful comment regarding the proportion of activated CAR-T cells in the DGLA-PAT CAR design. As noted, Fig. 3c, e, f reflect results from the DGLA-PAT CAR T system, in which PAT CAR expression is directly induced in T cells via light- and

tamoxifen-dependent transcription, without involvement of the synNotch mechanism. The observed CAR expression (~15-20%) and subsequent activation (~20%) are influenced by several factors. First, the efficiencies of tamoxifen-induced nuclear translocation and light-induced pMag-nMag dimerization are inherently variable in primary human T cells, and may constrain the overall induction rate (Allen et al., ACS Synth Biol, 2019). Second, the induction of large genetic cargos, such as the CAR construct, tends to be less efficient compared to smaller reporters like GFP, as we have observed in previous studies (Wu et al., Nat Biomed Eng, 2021). Consistently, the induction efficiency of the PEbody CAR (~1200 bp) is higher than that of CD19CAR (~1600 bp) (**Fig. 3e** vs **3c**). Third, expression levels of exogenous optogenetic components (e.g., pMag and nMag) are tightly regulated in T cells, further limiting the maximal response to light stimulation (Huang et al., Sci Adv, 2020).

To address these limitations, we later implemented the DGLA-sPAT CAR T design, in which tamoxifen and light are used to induce antigen expression in tumor cells rather than directly driving CAR expression (**Fig. 4a,b**). This approach achieves a much higher antigen induction efficiency (~86%, partly attributed to the small tCD19 size, as shown in **Fig. 4d,e**), which then activates PEbody PAT CAR expression in T cells via a synNotch mechanism with substantially improved efficiency (**Fig. 4f**). The modularity and scalability of the DGLA-sPAT design allow for better control and higher activation efficiency in engineered CAR-T cells.

3. For the concept of “training center” tumor, the initial step is T cells expressing synNotch recognize either endogenous or DGLA-induced antigen, which means CAR-T cell activity is still dependent on one antigen, and antigen loss or downregulation will likely make such CAR-T therapy nonresponsive. Can the authors please address (defend or refute this assertion) this important issue either experimentally or in text.

Response: We thank you for raising this important point regarding potential antigen loss and its impact on CAR-T responsiveness in the “training center” tumor strategy. In the DGLA-sPAT CAR T design, the expression of the tumor antigen that serves as the input for synNotch activation is not reliant on endogenous antigen expression, regulation, or evasion, but rather induced synthetically via the DGLA system, which is activated by tamoxifen and light with high efficiency (as shown in **Fig. 4d,e**). Once triggered, this induction results in stable and constitutive expression of the synthetic antigen on the tumor cell surface.

Furthermore, the DGLA gene cassette is stably integrated in the genome of tumor cells at a much higher copy number than that of typical endogenous antigen genes. This makes antigen loss or downregulation much more unlikely, thereby mitigating a major cause of resistance commonly seen in conventional CAR-T therapies. Since this is a synthetic and controllable gene cassette, we can also integrate genomic insulators flanking the inducible gene in the future for the stable maintenance of synthetic gene expression (Cabrera et al., Cell Syst, 2022).

Taken together, the DGLA-sPAT CAR T system is designed to be resilient against antigen escape, as the “training center” antigen is not subjected to the selective pressures that typically drive immune evasion through loss of endogenous targets. We have added this part in Discussion of the revised manuscript (line 341-347).

4. In Fig 6, it seems that a clear interpretation of these data would require evaluating the effects of blue light treatment and AAV infection in one tumor side, control groups without sPAT CAR T cells.

Response: We thank you for raising this important point. To directly assess the potential effects of blue light exposure and AAV delivery independent of sPAT CAR T cell activity, we performed an additional in vivo control study using a bilateral tumor model (**Fig. S9**). In this study, one tumor received both blue light illumination and AAV administration, while the contralateral tumor received no treatment (**Fig. S9a,b**). No sPAT CAR T cells were administered in this experiment. The results showed no significant difference in tumor growth between the treated and untreated sides (**Fig. S9c,d**), indicating that blue light and AAV delivery alone do not induce tumor regression. These findings, included in line 270-272 of the revised manuscript, support the conclusion that the therapeutic effects observed in Fig. 6 are specifically attributable to sPAT CAR T cell activity, rather than nonspecific effects of the light or vector delivery.

5. Overall, the in vivo CAR T immunotherapy studies need to be strengthened. Although experiments described in Fig6 showed that DGLA-PAT CAR strategy could work to initiate tumor control based on inducible tCD19 expression and anti-PD-L1 treatment, there is no evidence that DGLA-PAT CAR is better than traditional CAR-T strategy in tumor control. And although the concept is novel, there is concern that given that the DGLA-PAT CAR strategy is more complicated in cell engineering, stricter in tumor killing initiation, possibly resulting in

less tumor-fighting CAR T cells, a parallel comparison between DGLA-PAT CAR and traditional CAR is recommended. Can the authors please address (defend or refute this assertion) this important issue either experimentally or in text.

Response: We appreciate your comment regarding the need to evaluate the DGLA-PAT CAR strategy in comparison to traditional CAR-T therapies. Compared to traditional CAR-T approaches, the DGLA-sPAT CAR design offers two major advantages.

(1) Spatially confined CAR-T activation. Traditional CAR-T therapies often suffer from on-target/off-tumor toxicity, especially in solid tumors where target antigens may also be expressed at low levels on healthy tissues (Morgan et al., Mol Ther, 2010; Flugel et al., Nat Rev Clin Oncol, 2023). Particularly, in a bilateral model, one side representing the target tumor and the other mimicking normal tissue with a similar antigen profile, traditional CAR-T cells injected at the target tumor side still trafficked to and killed the distal tumor mimicking normal tissue expressing similar antigens, modeling potential off-tumor effects (Wu et al., Nat Biomed Eng, 2021). In contrast, the DGLA-sPAT CAR system enables precise spatial control of cytotoxicity via light-directed local induction of synthetic tumor antigens (**Fig. 5c-e**), allowing T cell activation confinement strictly within illuminated tumor regions (**Fig. 5f, Fig. 6**). This spatial gating offers a path to improved safety of DGLA-sPAT CAR over the traditional CAR T therapy.

(2) Antigen flexibility to overcome tumor heterogeneity and escape. Traditional CARs are hardwired to recognize a single predefined antigen, which can lead to tumor relapse due to antigen heterogeneity or antigen loss. Our strategy enables dynamic antigen control, allowing CAR-T cells to be reprogrammed to respond to different or multiple antigens either simultaneously or sequentially introducing corresponding antibodies. This capability, as demonstrated in **Fig. 1c,f** (antigen switching), **Fig. 2d,e** (tumor heterogeneity), and **Fig. 2c and Fig. 3h** (antigen escape), provides a modular platform to adapt CAR-T function in response to evolving tumor phenotypes.

In summary, compared with traditional CAR-T approaches, DGLA-sPAT CAR offers two key advantages: light-gated spatial confinement of cytotoxicity to reduce on-target/off-tumor toxicity and programmable multi-antigen control to address heterogeneity and antigen escape. We have emphasized this point in Discussion in the revised manuscript (line 318-321).

6. Only one tumor type, the MDA cell line, was used for in vivo tumor therapy study. To support the concept of “versatile” therapy strategy, multiple tumor types, antigens and antibodies need to be tested in more than one tumor model in vivo.

Response: We appreciate your suggestion regarding the need to demonstrate the versatility of our therapeutic strategy across multiple tumor models. To this end, we have conducted an additional in vivo experiment using a different tumor type: the prostate cancer cell line PC-3 with prostate-specific membrane antigen PSMA (**Fig. S5**). In this model, NSG mice were inoculated bilaterally with the engineered PC-3 cells with DGLA-inducible tCD19 (**Fig. S5a**). After systemic administration of sPAT CAR T cells via tail vein injection, we delivered an anti-PSMA-PE antibody to guide CAR-T killing (**Fig. S5b**). Consistent with our previous findings, the DGLA-sPAT CAR T system effectively controlled tumor growth, demonstrating high antigen specificity and minimal off-target effects (**Fig. S5c,d**). Importantly, the contralateral tumors, representing tissues with a similar antigen expression profile but without DGLA-induced target antigen presentation, showed no significant regression. These results support the generalizability and adaptability of our approach to different tumor types, antigens, and antibody-PE conjugates, further reinforcing the versatility of the DGLA-sPAT CAR T platform. We have included this result (line 247-249) and updated the Materials and Methods (line 520-536) in the revised manuscript.

7. The description of the results is not clear. In the spirit of rigor and transparency, it is critical that experimental details be provided.

Response: We thank you for emphasizing the importance of clarity and transparency in data presentation. In response, we have revised the manuscript to include more and substantial experimental details in Results and Materials and Methods sections. These updates clarify key aspects of the study design, experimental procedures, and data interpretation to ensure rigor and reproducibility. We believe these additions will help readers more clearly understand and evaluate the findings presented. Particularly, we have provided more details for the experiments below:

Activation markers and cytokine measurements (line 441-449)

In vitro killing assays (line 451-461)

CRISPR-Cas9 knockout of CD19 gene (line 463-472)

Antigen escape model experiments (line 474-477)

Tumor migration and infiltration assays (line 491-498)

Liver and kidney functional tests (line 500-504)

In vivo T cell homing and activation (line 520-536)

Ex vivo flow cytometry and frozen-section histology/IHC (line 538-558)

Minor concerns,

1. The surface expression of PEbody PAT CAR in T cells is not confirmed. This is important to know as this will impact signaling and ultimately efficacy.

Response: We thank you for pointing out this important consideration. To confirm surface expression of the PEbody PAT CAR in T cells, we have performed **live cell imaging of transduced T cells** using EGFP as marker to verify the surface expression of PEbody CAR (**Fig. S1a**, line 88 in the revised manuscript). The results clearly demonstrate that the PEbody PAT CAR is **expressed on the T cell surface**, supporting its proper localization and potential for antigen recognition and signaling.

2. In Fig1f, it seems that MCF-7 is not sensitive to killing by anti-HER2 and anti-MUC1. Please elaborate in the discussion.

Response: We thank you for the opportunity to further clarify our findings regarding MCF-7 cell sensitivity to PEbody CAR-mediated killing. In our system, the PEbody CAR T cells utilize antibodies to bridge between the CAR and tumor antigens such as HER-2 and MUC1 on the target cells. The efficacy of this approach is highly dependent on the expression level and conformation of these antigens on tumor cells. MCF-7 cells are well documented to express low levels of HER-2 (Slamon et al., Science, 1987; Holliday et al., Breast Cancer Res, 2011). Consequently, the low HER-2 density on MCF-7 cells results in fewer available binding sites for the anti-HER2 adapter, thereby reducing the effective engagement of the PEbody CAR T cells. Similarly, although MCF-7 cells do express MUC1, the antibody adapter's ability to mediate effective CAR T cell activation could be influenced by the antigen's conformation and glycosylation status. Gendler provided an in-depth analysis of MUC1 structure and highlighted how its glycosylation can modulate antibody binding (Gendler, J Mammary Gland Biol

Neoplasia, 2001), while epitope accessibility and post-translational modifications of MUC1 have been reported to affect the efficacy of antibody-mediated targeting (Movahedin et al., Glycobiology, 2017). Thus, the suboptimal presentation of MUC1 on MCF-7 cells likely contributes to the reduced cytotoxic response observed with anti-MUC1 adapter-mediated targeting.

In summary, the adapter-based mechanism of our PEbody CAR T cells is critically dependent on both the density and conformational state of the target antigens. The inherently low HER-2 expression and the less favorable presentation of MUC1 on MCF-7 cells explain the limited cytotoxicity observed in our experiments. We have incorporated these points in Discussion (line 349-362) to provide a clearer rationale for the observed differential sensitivity of MCF-7 cells.

3. Information of the antibodies used are not provided.

Response: We have now included detailed information for the antibodies used in Materials and Methods section, including clone names, sources, and catalog numbers to ensure clarity and reproducibility.

Line 442-443:

PE-conjugated anti-human CD19 antibody (clone HIB90, mouse IgG1, κ , BioLegend 302208)

Line 446-448:

APC-conjugated anti-human CD69 antibody (clone FN50, mouse IgG1, κ , Biolegend 310910)

APC-conjugated anti-human CD25 antibody (clone BC96, mouse IgG1, κ , Biolegend 302609)

Line 455-460:

CD19 (clone HIB19, mouse IgG1, κ , BioLegend 302208)

CD20 (clone 2H7, mouse IgG2b, κ , BioLegend 302306)

CD38 (clone HB-7, mouse IgG1, κ , BioLegend 356604)

MUC1 (clone 16A, mouse IgG1, λ , BioLegend 355604)

HER2 (clone 24D2, mouse IgG1, κ , BioLegend 324406)

PD-L1 (clone 29E.2A3, mouse IgG2b, κ , BioLegend 329706)

PSMA (LNI-17, mouse IgG1, κ , BioLegend 342504)

Line 468:

CD19 antibody for flow cytometry (clone HIB19, mouse IgG1, κ , Biolegend 302222)

Line 470:

Isotype control (mouse IgG1, κ , BioLegend 400130)

Line 532-535:

Anti-human PD-L1 antibody (clone 29E.2A3, mouse IgG2b, κ , BioLegend 329706)

Anti-human PSMA antibody (clone LNI-17, mouse IgG1, κ , BioLegend 342504)

Line 543-545:

Anti-human CD3-APC (clone OKT3; mouse IgG2a, κ ; BioLegend 317317)

Line 553: Rat anti-CD19 (clone 6OMP31; Thermo Fisher 53-0194-82; 1:500)

4. In Fig2c, details regarding the generation of 25%,50%,100% CD19 loss cell lines are not required.

Response: Briefly, the 25%, 50%, 75%, and 100% CD19 loss cells were generated by mixing CD19-knockout Nalm-6 cells (0% CD19+) with wild-type CD19+ Nalm-6 cells (100% CD19+) at the defined ratios of cell number to simulate varying levels of antigen escape. We have included these details in Materials and Methods (line 474-477).

5. In Fig3 and the verbiage in line 127-144 appears to be irrelevant to the topic of this manuscript. Also, although interesting, it is not clear how plausible it is for a CAR T therapy to engineer T cells with four transgenes for inducible CAR expression by Tam and light.

Response: The intent of Fig. 3 and lines 127-144 is to introduce and characterize the drug-gated light activation (DGLA) safety module, which we use to mitigate potential off-tumor cytotoxicity of the PAT CAR therapy. We have revised the text to make the contents more aligned with the topic of the manuscript (line 128-144).

Indeed, as the reviewer pointed out, the overall transduction efficiency of DGLA-PAT CAR T cells was low (~10-20 %), which caused difficulties for in vivo studies using this design.

Therefore, we have adopted a second design, the DGLA-sPAT CAR T (**Fig. 4**), to demonstrate the in vivo therapeutic effect of DGLA and PAT CAR. In this design, the DGLA-related transgenes are specifically delivered into tumor cells which can achieve high efficiency (~85% induction rate with the small size of tCD19 antigen, **Fig. 4e**), while T cells are engineered with synNotch and the PAT CAR reporter cassette, similar to the conventional synNotch design of which the potential for CAR T therapy has been well-demonstrated by previous studies (Morsut et al., Cell, 2016; Roybal et al., Cell, 2016).

6. In Fig6, direct evidence for in situ AAV-injection and light-induced tCD19 expression in MDA tumor is not provided (although the concept is interesting). The PEbody CAR-T cell infiltration in tumor has not been thoroughly characterized nor has their function.

Response: We appreciate your constructive suggestions. In the revised manuscript we have included immunohistochemistry study confirming tCD19 expression in AAV-injected, light-stimulated MDA tumors (**Fig. S8c**, line 264). This directly demonstrates successful intratumoral AAV transduction and light-dependent induction of the tCD19 target used for PEbody CAR engagement.

PEbody CAR-T cell infiltration: we previously showed tumor infiltration of PEbody CAR-T cells following tail-vein delivery using a PEbody CAR and firefly luciferase reporter cassette (Gal4UAS-Pmin-PEbody PAT CAR-P2A-Fluc, **Fig. 5c-e** and **Fig. S4b** in revised manuscript). In the revision, we further quantified intratumoral human T cells by flow cytometry of dissociated tumor tissue using anti-human CD3 antibody and PE staining, confirming PEbody CAR-T cell presence within the tumor microenvironment (**Fig. S6b**).

Functional characterization: the cytotoxic function of PEbody CAR-T cells is presented in **Fig. 1c-f** and **Fig. S1b-d**. We have performed more characterization of PEbody CAR T cells including up-regulation of activation markers CD25 and CD69 (**Fig. S1f**) and secretion of cytokines IFN- γ , TNF- α , and IL-2 (**Fig. S1g**) upon target antigen engagement.

Collectively, these additions provide direct validation of in situ AAV delivery and light-induced tCD19 expression and infiltration of PEbody CAR-T cells, and substantiate their functional activity.

Reviewer #3 (Remarks to the Author)

This is a timely article reporting the development of a monobody based PE targeting CAR (PAT) that allows for T cell reactivity to be directed to tumor antigens via PE-conjugated antibodies. The authors demonstrate the ability of PAT T cells to kill tumor cells in a specific manner based on the presence of PE-conjugated antibodies. They further improve the CAR safety profile by engineering the PAT under a light/tamoxifen controlled Cre recombinase (DGLA-PAT CAR), which restricts CAR activity to local illuminated areas and requires the presence of tamoxifen. The authors also engineered a synNotch regulated PAT (sPAT), whereby the PAT expression is

induced by priming of the SynNotch antigen. Most notably, the authors devised a CAR-antigen pairing system, whereby tumor antigen expression is controlled by the DGLA system and T cell reactivity is controlled by the sPAT modules (DGLA-sPAT). In this strategy, tumor cells can be vaccinated with an inducible antigen that can then “train” T cells to react specifically at the tumor site through the synNotch module. In murine flank models of human breast cancer, the authors show MDA cells engineered or vaccinated with the DGLA inducible CD19 cassette, express the target antigen only under illumination and with tamoxifen treatment. Administration of sPAT T cells and PE-conjugated PD-L1 antibody resulted in tumor regression only in tumors that were illuminated with light. These efforts are framed in the context of antigen escape, tumor heterogeneity, and safety concerns arising from clinical experience of standard CAR-T therapy, particularly for solid tumors. The impact and novelty of the manuscript is well suited for Nature Communications. The following revision experiments/edits would strengthen the manuscript:

1. In cytotoxicity assays (Figure 1 and Figure 3), the authors should include a conventional CAR control as a benchmark. It is unclear whether the inducible CAR formats reported in this study are of similar potency as standard CARs used in the field. The authors do a good job of motivating the clinical relevance of the study (i.e. addresses antigen heterogeneity, safety, and the biocompatibility of PE), but having a benchmark for potency is also needed to assess the clinical relevance. In addition to tumor killing, measuring the levels of cytokine secretion (i.e. IFN γ , IL-2) from tumor cocultures is appropriate for assessing T cell functionality.

Response: We really appreciate your suggestion. We agree that benchmarking against a conventional CAR and assessing cytokine secretion are important for evaluating clinical relevance. In the revised manuscript, we have added conventional CAR controls to the cytotoxicity assays (**Fig. 1c,e** and **Fig. 3d,g** in the revised manuscript). We also included ELISA measurements of key secreted cytokines including IFN- γ , TNF- α , and IL-2 after tumor co-cultures (**Fig. S1g**), providing functional readouts complementary to target-cell killing.

2. Similar to comment 1, Mock (untransduced) T cells are the appropriate negative controls in cytotoxicity assays that allow the reader to assess the leaky or non-specific activity of the system and should be included.

Response: Mock controls using untransduced T cells have been included in the cytotoxicity assays (**Fig. 1c,e** and **Fig. 3d,g** in the revised manuscript) and noted in figure legend.

3. In Figure 3C, are the cells gated on transgene+? It is unclear if the relatively low induction of CD19 CAR-T (~15%) is because of inefficiencies of the inducible system or if it is due to low transduction efficiencies. The authors should clarify these details.

Response: We thank you for the suggestion. For the assays in Fig. 3c, we used FACS-sorted T cells containing the transgenes of the inducible system. Thus, the result reflects transgene positive cells rather than variability from mixed transduction efficiencies. The ~15% induction likely arises from limited expression of the DGLA regulator components in primary T cells, where transgene expression is tightly regulated. This phenomenon is consistent with our previous findings, where we observed that the efficiency of induction is dependent on regulator expression levels in T cells (Huang et al., Sci Adv, 2020). To address this limitation, we subsequently adopted an alternative strategy where the DGLA system is delivered into tumor cells rather than T cells (**Fig. 4a**), which allows for stronger expression of tCD19 antigen and significantly improved induction efficiency (~85%, **Fig. 4e**).

4. The cytotoxicity data presented in this study are normalized. The authors should indicate what the plots were normalized to.

Response: We appreciate your suggestion for clarification. The cytotoxicity data were normalized to tumor-only control wells without T cells. Details regarding this normalization method have been added to the figure legends and Materials and Methods section in the revised manuscript (line 460-461).

5. In Figure 5E, it is unclear if the increase in fluc signal is driven by T cell expansion or synNotch activation since there is no constitutive and orthogonal luciferase that was coexpressed in the T cells. Did the authors administer PE conjugated antibody in this experiment? Adequate clarification or additional experiments should be provided to help interpret the source of the increasing fluc signal.

Response: We thank you for this insightful question. In the experiment shown in Figure 5E, PE-conjugated antibody was administered one day after T cell injection. We believe that the initial

Fluc signal observed is primarily driven by synNotch activation of the injected T cells homing to the tCD19+ tumor site, rather than T cell expansion per se. This is because T cell expansion would require activation signaling, which in our system depends on the synNotch-mediated induction of PEbody CAR followed by its engagement with the target antigen via the corresponding PE-conjugated antibody. It is also important to note that in our synNotch-inducible Fluc design, any newly-generated T cells (resulting from expansion) would still require synNotch activation to express Fluc. Therefore, while T cell expansion may contribute to the overall signal increase over time, T cell expansion alone may not increase the Fluc signal without the synNotch activation. We have clarified this point in the revised manuscript (line 224-225) to ensure accurate interpretation of these results.

6. The authors should add a reference for the following study that demonstrated an inducible CAR regulated by tamoxifen activated recombinase.

Chakravarti, Deboki, et al. "Inducible gene switches with memory in human T cells for cellular immunotherapy." ACS synthetic biology 8.8 (2019): 1744-1754.

Response: We thank you for bringing this relevant reference to our attention. We have added this reference in the revised manuscript (line 279-280) to acknowledge this prior work on inducible CAR regulation.

References

1. Allen, M. E., et al. An AND-Gated Drug and Photoactivatable Cre-loxP System for Spatiotemporal Control in Cell-Based Therapeutics. *ACS Synth Biol* **8**, 2359-2371 (2019).
2. Cabrera, A., et al. The sound of silence: Transgene silencing in mammalian cell engineering. *Cell Syst* **13**, 950-973 (2022).
3. Chappaz, S. and D. Finke The IL-7 signaling pathway regulates lymph node development independent of peripheral lymphocytes. *J Immunol* **184**, 3562-3569 (2010).
4. Di Stasi, A., et al. Inducible apoptosis as a safety switch for adoptive cell therapy. *N Engl J Med* **365**, 1673-1683 (2011).
5. Flugel, C. L., et al. Overcoming on-target, off-tumour toxicity of CAR T cell therapy for solid tumours. *Nat Rev Clin Oncol* **20**, 49-62 (2023).
6. Gendler, S. J. MUC1, the renaissance molecule. *J Mammary Gland Biol Neoplasia* **6**, 339-353 (2001).
7. Holliday, D. L. and V. Speirs Choosing the right cell line for breast cancer research. *Breast Cancer Res* **13**, 215 (2011).
8. Huang, Z., et al. Engineering light-controllable CAR T cells for cancer immunotherapy. *Sci Adv* **6**, eaay9209 (2020).

9. Lee, S. Y., et al. CD8(+) chimeric antigen receptor T cells manufactured in absence of CD4(+) cells exhibit hypofunctional phenotype. *J Immunother Cancer* **11** (2023).
10. Morgan, R. A., et al. Case report of a serious adverse event following the administration of T cells transduced with a chimeric antigen receptor recognizing ERBB2. *Mol Ther* **18**, 843-851 (2010).
11. Morsut, L., et al. Engineering Customized Cell Sensing and Response Behaviors Using Synthetic Notch Receptors. *Cell* **164**, 780-791 (2016).
12. Movahedin, M., et al. Glycosylation of MUC1 influences the binding of a therapeutic antibody by altering the conformational equilibrium of the antigen. *Glycobiology* **27**, 677-687 (2017).
13. Park, A. K., et al. Effective combination immunotherapy using oncolytic viruses to deliver CAR targets to solid tumors. *Sci Transl Med* **12** (2020).
14. Roybal, K. T., et al. Engineering T Cells with Customized Therapeutic Response Programs Using Synthetic Notch Receptors. *Cell* **167**, 419-432 e416 (2016).
15. Slamon, D. J., et al. Human breast cancer: correlation of relapse and survival with amplification of the HER-2/neu oncogene. *Science* **235**, 177-182 (1987).
16. Terakura, S., et al. Generation of CD19-chimeric antigen receptor modified CD8+ T cells derived from virus-specific central memory T cells. *Blood* **119**, 72-82 (2012).
17. Wu, Y., et al. Control of the activity of CAR-T cells within tumours via focused ultrasound. *Nat Biomed Eng* **5**, 1336-1347 (2021).

We thank the reviewers for their thorough evaluation of our manuscript and their constructive feedback. We have conducted additional experiments extensively and addressed all comments and suggestions, which have helped us improve the clarity and quality of our work. Below, we provide detailed responses to each reviewer's comments and describe the revisions made to address their concerns. We believe these changes have significantly strengthened the manuscript.

Reviewer #1 (Remarks to the Author)

In this study, the authors developed a novel CAR that can be programmed with PE-conjugated antibodies to target multiple user-defined antigens, which allows for overcoming antigen escape and tumor heterogeneity. In this CAR-T system, by integrating PAT CAR and DGLA gene expression, DGLA-PAT CAR can confine the CAR T activation in the local tumor tissue and reduce off-target toxicity. this is a meaningful study. However, there are some experiments and controls that detract from the significance of this work. The following suggestions also should be taken into account.

1. The authors emphasize the local infiltration and activation of CAR-T and speculate that the expression of CAR-T diminishes after it migrates out of the tumor tissue. In animal models, it is recommended to supplement data related to the detection of CAR-T in peripheral blood, spleen, and draining lymph nodes after CAR-T treatment. In addition, it is recommended to add safety-related studies, such as liver and kidney function tests, as well as pathologic evaluation of important organs.

Response: We appreciate this constructive suggestion and have added new data on CAR-T cell detection and safety-related studies. Briefly, we performed detection of DGLA-PAT CAR-T cells in peripheral blood and spleen after CAR-T treatment (**Fig. S6a**). The draining lymph nodes cannot be located due to their under development in NSG mice (Chappaz et al., J Immunol, 2010). The non-activated T cells, via anti-human CD3 antibody staining, are readily detectable in these locations, but none of them were activated as shown by the negative signals in PAT-CAR staining (**Fig. S6a**). Consistent with our design, the DGLA-PAT CAR-T cells were significantly activated in tumor region evidenced by the clear PAT CAR staining (**Fig. S6b**), confirming the intended tumor-specific activation.

Histopathology study, via H&E staining, of heart, lung, spleen, kidney, liver, and brain after CAR-T treatment (**Fig. S7a**), as well as liver and kidney functional tests were further performed (**Fig. S7b**). No treatment-related lesions or toxicities are observed. These results verified the safety of our approach. The Materials and Methods (line 555-575) and Results (line 249-254) sections have been updated accordingly.

2. In Figure 2b, Isotype as a control should be added to verify the knockdown effect of CD19 in Nalm-6 cells.

Response: Thank you for pointing this out. We have now included the isotype control staining result to verify the effect of CD19-knockout in the Nalm-6 CD19-KO cell line (**Fig. S2a**, line 112-113). The isotype control profile has also been included in the revised **Fig. 2B** (CD19 staining histogram), and noted in the figure legend.

3. CD19 is a lineage marker for B cells, and the authors applied CD19+/PSMA-, as well as CD19-/PSMA+ prostate cancer PC-3 cells in their experiments, which required the detection of CD19 as well as PSMA expression in PC-3 cells

Response: We have now validated the CD19 and PSMA antigen profiles of the engineered PC-3 cell lines and included the data in the revised manuscript (**Fig. S2b-c**, line 122-123).

4. In Figure 3, the authors constructed Tam-light induced PAT CAR-T and verified the ability to target kill target cells. However, the result does not seem to be sufficiently confirmed to attenuate the off-tumor toxic response to CAR-T therapy.

Response: We appreciate your insightful feedback regarding the evaluation of off-tumor toxicity attenuation in our study. In response, we would like to clarify two important points:

First, we have demonstrated in **Fig. 3g-h** that the cytotoxicity of DGLA PAT CAR-T cells can be effectively regulated through external stimuli such as light. These results indicate that the cytotoxicity of the engineered DGLA CAR-T cells can be precisely confined at the tumor site, supporting the potential for reducing off-tumor toxicity through on-demand activation such as light stimulation on tumor regions as shown in **Fig. 5c-f** and **Fig. 6**.

Second, due to the low efficiency of delivering all the three gene cassettes into primary T cells, it would be difficult to perform a direct *in vivo* validation with this initial design. Recognizing

these limitations, we subsequently transitioned to an alternative design that is more suitable for in vivo evaluation (**Fig. 4a-b**), as detailed in the later sections of our manuscript (**Fig. 5c-f, Fig. 6, and Fig. S10**), which demonstrated attenuated off-tumor toxic response to CAR-T therapy.

5. Figure S3b, Lack of statistical analysis results for left and right side tumors before treatment. From the results of Day15, before treatment, the left side tCD19+ MDA tumors were smaller than the right side tCD19- MDA tumors, and the results of pre-treatment statistical analysis were required to ensure the reliability of the results. In addition, considering the possible effects of gene editing on MDA-MD-231 cells, the effects of expressing tCD19 on the proliferation, invasion and migration of MDA-MD-231 need to be verified.

Response: We appreciate the reviewer's concern regarding potential baseline differences in tumor size prior to CAR-T-cell treatment. We have performed the statistical analysis on tumor growth comparing the left and right sides, each normalized to their respective Day 7 measurements, to evaluate any pre-treatment imbalance. As shown in **Fig. 5f** (associated with IVIS images of **Fig. S4c** in the revised manuscript), there was no statistically significant difference between the left side (tCD19+) and right side (tCD19-) MDA-MB-231 tumors prior to CAR-T-cell injection (noted as "ns" in **Fig. 5f**, $p > 0.05$). A significant divergence emerged only after CAR-T treatment, confirming that the therapeutic effect, rather than an initial size bias, accounts for the observed difference on tumor growth.

Regarding the potential effects of gene editing and tCD19 expression on MDA-MB-231 cell behavior, we respectfully note that tCD19 is a truncated, non-signaling surface marker that has been extensively validated as a clinically safe selection marker (Di Stasi et al., N Engl J Med, 2011; Terakura et al., Blood, 2012; Lee et al., J Immunother Cancer, 2023). It lacks intracellular signaling domains and has no known impact on cell proliferation, migration, or invasion. This inert nature has been demonstrated in multiple cell types, including T cells and hematopoietic stem cells, without affecting their biological behavior (Park et al., Sci Transl Med, 2020). We have also performed a comparison of proliferation, migration, and invasion between tCD19+ and tCD19- MDA-MB-231 lines, and observed no significant difference (**Fig. S4a**, line 217-218). Therefore, the use of tCD19 in our study is unlikely to alter the intrinsic properties of MDA-MB-231 cells.

6. In figure 3h, the results showed a difference in cytotoxicity between the non-induced and induced groups in CD19- Nalm-6 cells (100% loss of CD19 antigen). However, the non-induced group were normal activated T cells and the induced group were CAR-T cells that could target CD19. It is theoretically implausible that there is a difference in cytotoxicity between these two groups in CD19- cells.

Response: We thank you for this critical note. As correctly noted, CD19- Nalm-6 cells (with complete loss of CD19 expression) should not exhibit differential sensitivity to non-induced T cells versus induced PEbody CAR-T cells, as the antigen is absent. The small but statistically significant difference observed in the initial experiment is possibly attributable to experimental variation rather than true antigen-specific cytotoxicity. To clarify this point, we have re-conducted this assay with more biological repeats ($n = 4$). The updated results, now included in the revised **Fig. 3h**, show no significant difference between these two groups, consistent with the PEbody CAR design.

7. The activation of CAR-T cells requires more detailed characterization, such as activation markers and cytokines.

Response: We thank you for this important suggestion. To provide a more comprehensive characterization of CAR-T cell activation, we have performed additional experiments to characterize surface activation markers and cytokine release. Specifically, we analyzed the expression of CD69 and CD25 by flow cytometry (**Fig. S1f**), and measured the secretion of key cytokines including IFN- γ , TNF- α , and IL-2 via ELISA (**Fig. S1g**). These results further confirm the antigen-specific activation of CAR-T cells and are now included in the revised manuscript (line 101-103).

Reviewer #2 (Remarks to the Author)

This manuscript from Ziliang Huang and colleagues described a strategy of T cell engineering for cancer therapy, which combines the applications of a PE-binding protein domain, the synNotch-regulated gene expression system, the tamoxifen-induced protein translocation system, and the light-responsive protein dimerization system. The authors aim to use the presented system to solve the current problems in T cell-mediated cancer therapy including the antigen loss, tumor heterogeneity, off-tumor toxicity, etc. They clearly showed that their DGLA-sPAT

CAR system could work in experimental models based on CD19, PD-L1 as model antigens. Below are concerns that if addressed could expand the significance of the studies and offer clarification on certain points.

1. PEbody is generated by directed evolution. It's not clear whether PEbody can also bind cell/tissue in host, which means using it in a CAR might be risky for an on-target/off-PE effect. If this is the first case of using it in a CAR, there could great value in investigating this attribute in the present study.

Response: We thank you for highlighting this important point. In the current work, we have assessed the binding specificity of PEbody when used as the antigen binding motif in CAR against several human cell lines originated from different tissues including blood (Nalm-6, **Fig. 1c**), breast (MCF-7, MDA-MB-231, **Fig. 1f**), and prostate (PC-3, **Fig. 1f**). Our results showed that when PE-conjugated antibody is absent, there is no binding/killing against the target cells, indicating the binding of PEbody towards host cells/tissues is minimal. To further address the potential concern of “on-target/off-PE” effects in vivo, we performed histological studies of important organs, including heart, lung, spleen, kidney, liver, and brain (**Fig. S7a**), as well as liver and kidney functional assays (**Fig. S7b**), after PEbody PAT CAR T treatment. Our results revealed no signs of tissue damage or inflammation, supporting the conclusion that PEbody CAR-T cells do not exhibit detectable off-PE toxicity in host tissues. These new results have been included in the revised manuscript (line 252-254).

2. In the DGLA-PAT CAR design, tumor antigen is induced by light and Tamoxifen, PAT CAR expression is induced by synNotch recognition of induced tumor antigen, PAT CAR-T killing is induced by PE-antibodies. The concern is that due to multiple gatekeeping events, the eventual percentage of cancer-fighting CAR-T cells seems to be on the lower side of transferred CAR-T cells. According to Fig3c, e, f, the CAR induction is around 15-20%, activation is around 20%. Can the authors articulate or speculate as to why only a subset of these CARs are activated?

Response: We appreciate your thoughtful comment regarding the proportion of activated CAR-T cells in the DGLA-PAT CAR design. As noted, Fig. 3c, e, f reflect results from the DGLA-PAT CAR T system, in which PAT CAR expression is directly induced in T cells via light- and tamoxifen-dependent transcription, without involvement of the synNotch mechanism. The

observed CAR expression (~15-20%) and subsequent activation (~20%) are influenced by several factors. First, the efficiencies of tamoxifen-induced nuclear translocation and light-induced pMag-nMag dimerization are inherently variable in primary human T cells, and may constrain the overall induction rate (Allen et al., ACS Synth Biol, 2019). Second, the induction of large genetic cargos, such as the CAR construct, tends to be less efficient compared to smaller reporters like GFP, as we have observed in previous studies (Wu et al., Nat Biomed Eng, 2021). Consistently, the induction efficiency of the PEbody CAR (~1200 bp) is higher than that of CD19CAR (~1600 bp) (**Fig. 3e** vs **3c**). Third, expression levels of exogenous optogenetic components (e.g., pMag and nMag) are tightly regulated in T cells, further limiting the maximal response to light stimulation (Huang et al., Sci Adv, 2020).

To address these limitations, we later implemented the DGLA-sPAT CAR T design, in which tamoxifen and light are used to induce antigen expression in tumor cells rather than directly driving CAR expression (**Fig. 4a,b**). This approach achieves a much higher antigen induction efficiency (~86%, partly attributed to the small tCD19 size, as shown in **Fig. 4d,e**), which then activates PEbody PAT CAR expression in T cells via a synNotch mechanism with substantially improved efficiency (**Fig. 4f**). The modularity and scalability of the DGLA-sPAT design allow for better control and higher activation efficiency in engineered CAR-T cells.

3. For the concept of “training center” tumor, the initial step is T cells expressing synNotch recognize either endogenous or DGLA-induced antigen, which means CAR-T cell activity is still dependent on one antigen, and antigen loss or downregulation will likely make such CAR-T therapy nonresponsive. Can the authors please address (defend or refute this assertion) this important issue either experimentally or in text.

Response: We thank you for raising this important point regarding potential antigen loss and its impact on CAR-T responsiveness in the “training center” tumor strategy. In the DGLA-sPAT CAR T design, the expression of the tumor antigen that serves as the input for synNotch activation is not reliant on endogenous antigen expression, regulation, or evasion, but rather induced synthetically via the DGLA system, which is activated by tamoxifen and light with high efficiency (as shown in **Fig. 4d,e**). Once triggered, this induction results in stable and constitutive expression of the synthetic antigen on the tumor cell surface.

Furthermore, the DGLA gene cassette is stably integrated in the genome of tumor cells at a much higher copy number than that of typical endogenous antigen genes. This makes antigen loss or downregulation much more unlikely, thereby mitigating a major cause of resistance commonly seen in conventional CAR-T therapies. Since this is a synthetic and controllable gene cassette, we can also integrate genomic insulators flanking the inducible gene in the future for the stable maintenance of synthetic gene expression (Cabrera et al., Cell Syst, 2022).

Taken together, the DGLA-sPAT CAR T system is designed to be resilient against antigen escape, as the “training center” antigen is not subjected to the selective pressures that typically drive immune evasion through loss of endogenous targets. We have added this part in Discussion of the revised manuscript (line 356-362).

4. In Fig 6, it seems that a clear interpretation of these data would require evaluating the effects of blue light treatment and AAV infection in one tumor side, control groups without sPAT CAR T cells.

Response: We thank you for raising this important point. To directly assess the potential effects of blue light exposure and AAV delivery independent of sPAT CAR T cell activity, we performed an additional in vivo control study using a bilateral tumor model (**Fig. S9**). In this study, one tumor received both blue light illumination and AAV administration, while the contralateral tumor received no treatment (**Fig. S9a,b**). No sPAT CAR T cells were administered in this experiment. The results showed no significant difference in tumor growth between the treated and untreated sides (**Fig. S9c,d**), indicating that blue light and AAV delivery alone do not induce tumor regression. These findings, included in line 270-272 of the revised manuscript, support the conclusion that the therapeutic effects observed in Fig. 6 are specifically attributable to sPAT CAR T cell activity, rather than nonspecific effects of the light or vector delivery.

5. Overall, the in vivo CAR T immunotherapy studies need to be strengthened. Although experiments described in Fig6 showed that DGLA-PAT CAR strategy could work to initiate tumor control based on inducible tCD19 expression and anti-PD-L1 treatment, there is no evidence that DGLA-PAT CAR is better than traditional CAR-T strategy in tumor control. And although the concept is novel, there is concern that given that the DGLA-PAT CAR strategy is more complicated in cell engineering, stricter in tumor killing initiation, possibly resulting in

less tumor-fighting CAR T cells, a parallel comparison between DGLA-PAT CAR and traditional CAR is recommended. Can the authors please address (defend or refute this assertion) this important issue either experimentally or in text.

Response: We appreciate your comment regarding the need to evaluate the DGLA-PAT CAR strategy in comparison to traditional CAR-T therapies. Compared to traditional CAR-T approaches, the DGLA-sPAT CAR design offers two major advantages.

(1) Spatially confined CAR-T activation. Traditional CAR-T therapies often suffer from on-target/off-tumor toxicity, especially in solid tumors where target antigens may also be expressed at low levels on healthy tissues (Morgan et al., Mol Ther, 2010; Flugel et al., Nat Rev Clin Oncol, 2023). Particularly, in a bilateral model, one side representing the target tumor and the other mimicking normal tissue with a similar antigen profile, traditional CAR-T cells injected at the target tumor side still trafficked to and killed the distal tumor mimicking normal tissue expressing similar antigens, modeling potential off-tumor effects (Wu et al., Nat Biomed Eng, 2021). In contrast, the DGLA-sPAT CAR system enables precise spatial control of cytotoxicity via light-directed local induction of synthetic tumor antigens (**Fig. 5c-e**), allowing T cell activation confinement strictly within illuminated tumor regions (**Fig. 5f, Fig. 6**). This spatial gating offers a path to improved safety of DGLA-sPAT CAR over the traditional CAR T therapy.

(2) Antigen flexibility to overcome tumor heterogeneity and escape. Traditional CARs are hardwired to recognize a single predefined antigen, which can lead to tumor relapse due to antigen heterogeneity or antigen loss. Our strategy enables dynamic antigen control, allowing CAR-T cells to be reprogrammed to respond to different or multiple antigens either simultaneously or sequentially introducing corresponding antibodies. This capability, as demonstrated in **Fig. 1c,f** (antigen switching), **Fig. 2d,e** (tumor heterogeneity), and **Fig.2c and Fig. 3h** (antigen escape), provides a modular platform to adapt CAR-T function in response to evolving tumor phenotypes.

In addition, we performed a direct head-to-head comparison between DGLA-sPAT CAR-T cells and conventional CAR-T cells using a PSMA⁺ tumor model (**Fig. S10a**), in which human PSMA was also expressed in the mouse liver via lentiviral transduction (**Fig. S10b-d**). This enables evaluation of both antitumor efficacy and on-target off-tumor toxicity (OTOT). As shown in **Fig.**

S10, both DGLA-sPAT CAR-T and conventional CAR-T cells efficiently controlled tumor growth (**Fig. S10e-g**), demonstrating that the DGLA-sPAT design does not compromise antitumor potency. However, mice receiving conventional anti-PSMA CAR-T cells exhibited marked systemic toxicity, including significant weight loss (**Fig. S10h**), elevated inflammatory cytokines (**Fig. S10i-k**), increased serum alanine aminotransferase levels (**Fig. S10l**), and pronounced lymphocytic infiltration with hepatocyte damage in the liver (**Fig. S10m**). In contrast, none of these OTOT-associated toxicities were observed in DGLA-sPAT-treated mice, whose organs remained histologically normal and showed no immune infiltration. Thus, our results demonstrate that the DGLA-sPAT CAR platform maintains potent tumor control while substantially improving safety through spatially restricted CAR activation, an advantage over traditional CAR-T approach in the revised manuscript (line 274-290).

In summary, compared with traditional CAR-T approaches, DGLA-sPAT CAR offers two key advantages: programmable multi-antigen control to address heterogeneity and antigen escape, and light-gated spatial confinement of cytotoxicity to reduce on-target/off-tumor toxicity. We have emphasized this point in Discussion in the revised manuscript (line 333-336).

6. Only one tumor type, the MDA cell line, was used for in vivo tumor therapy study. To support the concept of “versatile” therapy strategy, multiple tumor types, antigens and antibodies need to be tested in more than one tumor model in vivo.

Response: We appreciate your suggestion regarding the need to demonstrate the versatility of our therapeutic strategy across multiple tumor models. To this end, we have conducted an additional in vivo experiment using a different tumor type: the prostate cancer cell line PC-3 with prostate-specific membrane antigen PSMA (**Fig. S5**). In this model, NSG mice were inoculated bilaterally with the engineered PC-3 cells with DGLA-inducible tCD19 (**Fig. S5a**). After systemic administration of sPAT CAR T cells via tail vein injection, we delivered an anti-PSMA-PE antibody to guide CAR-T killing (**Fig. S5b**). Consistent with our previous findings, the DGLA-sPAT CAR T system effectively controlled tumor growth, demonstrating high antigen specificity and minimal off-target effects (**Fig. S5c,d**). Importantly, the contralateral tumors, representing tissues with a similar antigen expression profile but without DGLA-induced target antigen presentation, showed no significant regression. These results support the generalizability and

adaptability of our approach to different tumor types, antigens, and antibody-PE conjugates, further reinforcing the versatility of the DGLA-sPAT CAR T platform. We have included this result (line 247-249) and updated the Materials and Methods (line 537-553) in the revised manuscript.

7. The description of the results is not clear. In the spirit of rigor and transparency, it is critical that experimental details be provided.

Response: We thank you for emphasizing the importance of clarity and transparency in data presentation. In response, we have revised the manuscript to include more and substantial experimental details in Results and Materials and Methods sections. These updates clarify key aspects of the study design, experimental procedures, and data interpretation to ensure rigor and reproducibility. We believe these additions will help readers more clearly understand and evaluate the findings presented. Particularly, we have provided more details for the experiments below:

Activation markers and cytokine measurements (line 456-464)

In vitro killing assays (line 466-476)

CRISPR-Cas9 knockout of CD19 gene (line 478-487)

Antigen escape model experiments (line 489-492)

Tumor migration and infiltration assays (line 506-513)

Kidney and liver functional tests (line 515-521)

In vivo T cell homing and activation (line 537-553)

Ex vivo flow cytometry and frozen-section histology/IHC (line 555-575)

Minor concerns,

1. The surface expression of PEbody PAT CAR in T cells is not confirmed. This is important to know as this will impact signaling and ultimately efficacy.

Response: We thank you for pointing out this important consideration. To confirm surface expression of the PEbody PAT CAR in T cells, we have performed **live cell imaging of transduced T cells** using EGFP as marker to verify the surface expression of PEbody CAR (**Fig. S1a**, line 88 in the revised manuscript). The results clearly demonstrate that the PEbody PAT

CAR is **expressed on the T cell surface**, supporting its proper localization and potential for antigen recognition and signaling.

2. In Fig1f, it seems that MCF-7 is not sensitive to killing by anti-HER2 and anti-MUC1. Please elaborate in the discussion.

Response: We thank you for the opportunity to further clarify our findings regarding MCF-7 cell sensitivity to PEbody CAR-mediated killing. In our system, the PEbody CAR T cells utilize antibodies to bridge between the CAR and tumor antigens such as HER-2 and MUC1 on the target cells. The efficacy of this approach is highly dependent on the expression level and conformation of these antigens on tumor cells. MCF-7 cells are well documented to express low levels of HER-2 (Slamon et al., Science, 1987; Holliday et al., Breast Cancer Res, 2011). Consequently, the low HER-2 density on MCF-7 cells results in fewer available binding sites for the anti-HER2 adapter, thereby reducing the effective engagement of the PEbody CAR T cells. Similarly, although MCF-7 cells do express MUC1, the antibody adapter's ability to mediate effective CAR T cell activation could be influenced by the antigen's conformation and glycosylation status. Gendler provided an in-depth analysis of MUC1 structure and highlighted how its glycosylation can modulate antibody binding (Gendler, J Mammary Gland Biol Neoplasia, 2001), while epitope accessibility and post-translational modifications of MUC1 have been reported to affect the efficacy of antibody-mediated targeting (Movahedin et al., Glycobiology, 2017). Thus, the suboptimal presentation of MUC1 on MCF-7 cells likely contributes to the reduced cytotoxic response observed with anti-MUC1 adapter-mediated targeting.

In summary, the adapter-based mechanism of our PEbody CAR T cells is critically dependent on both the density and conformational state of the target antigens. The inherently low HER-2 expression and the less favorable presentation of MUC1 on MCF-7 cells explain the limited cytotoxicity observed in our experiments. We have incorporated these points in Discussion (line 364-377) to provide a clearer rationale for the observed differential sensitivity of MCF-7 cells.

3. Information of the antibodies used are not provided.

Response: We have now included detailed information for the antibodies used in Materials and Methods section, including clone names, sources, and catalog numbers to ensure clarity and reproducibility.

Line 457-458:

PE-conjugated anti-human CD19 antibody (clone HIB90, mouse IgG1, κ , BioLegend 302208)

Line 461-464:

APC-conjugated anti-human CD69 antibody (clone FN50, mouse IgG1, κ , Biolegend 310910)

APC-conjugated anti-human CD25 antibody (clone BC96, mouse IgG1, κ , Biolegend 302609)

Line 471-475:

CD19 (clone HIB19, mouse IgG1, κ , BioLegend 302208)

CD20 (clone 2H7, mouse IgG2b, κ , BioLegend 302306)

CD38 (clone HB-7, mouse IgG1, κ , BioLegend 356604)

MUC1 (clone 16A, mouse IgG1, λ , BioLegend 355604)

HER2 (clone 24D2, mouse IgG1, κ , BioLegend 324406)

PD-L1 (clone 29E.2A3, mouse IgG2b, κ , BioLegend 329706)

PSMA (LNI-17, mouse IgG1, κ , BioLegend 342504)

Line 484-485:

CD19 antibody for flow cytometry (clone HIB19, mouse IgG1, κ , Biolegend 302222)

Line 485-486:

Isotype control (mouse IgG1, κ , BioLegend 400130)

Line 560-562:

Anti-human CD3-APC (clone OKT3; mouse IgG2a, κ ; BioLegend 317317)

Line 570: Rat anti-CD19 (clone 6OMP31; Thermo Fisher 53-0194-82; 1:500)

4. In Fig2c, details regarding the generation of 25%,50%,100% CD19 loss cell lines are not required.

Response: Briefly, the 25%, 50%, 75%, and 100% CD19 loss cells were generated by mixing CD19-knockout Nalm-6 cells (0% CD19⁺) with wild-type CD19⁺ Nalm-6 cells (100% CD19⁺) at the defined ratios of cell number to simulate varying levels of antigen escape. We have included these details in Materials and Methods (line 489-492).

5. In Fig3 and the verbiage in line 127-144 appears to be irrelevant to the topic of this manuscript. Also, although interesting, it is not clear how plausible it is for a CAR T therapy to engineer T cells with four transgenes for inducible CAR expression by Tam and light.

Response: The intent of Fig. 3 and lines 127-144 is to introduce and characterize the drug-gated light activation (DGLA) safety module, which we use to mitigate potential off-tumor cytotoxicity of the PAT CAR therapy. We have revised the text to make the contents more aligned with the topic of the manuscript (line 128-144).

Indeed, as the reviewer pointed out, the overall transduction efficiency of DGLA-PAT CAR T cells was low (~10-20 %), which caused difficulties for in vivo studies using this design. Therefore, we have adopted a second design, the DGLA-sPAT CAR T (**Fig. 4**), to demonstrate the in vivo therapeutic effect of DGLA and PAT CAR. In this design, the DGLA-related transgenes are specifically delivered into tumor cells which can achieve high efficiency (~85% induction rate with the small size of tCD19 antigen, **Fig. 4e**), while T cells are engineered with synNotch and the PAT CAR reporter cassette, similar to the conventional synNotch design of which the potential for CAR T therapy has been well-demonstrated by previous studies (Morsut et al., Cell, 2016; Roybal et al., Cell, 2016).

6. In Fig6, direct evidence for in situ AAV-injection and light-induced tCD19 expression in MDA tumor is not provided (although the concept is interesting). The PEbody CAR-T cell infiltration in tumor has not been thoroughly characterized nor has their function.

Response: We appreciate your constructive suggestions. In the revised manuscript we have included immunohistochemistry study confirming tCD19 expression in AAV-injected, light-stimulated MDA tumors (**Fig. S8c**, line 264). This directly demonstrates successful intratumoral AAV transduction and light-dependent induction of the tCD19 target used for PEbody CAR engagement.

PEbody CAR-T cell infiltration: we showed tumor infiltration of PEbody CAR-T cells following tail-vein delivery using a PEbody CAR and firefly luciferase reporter cassette (Gal4UAS-Pmin-PEbody PAT CAR-P2A-Fluc, **Fig. 5c-e** and **Fig. S4b** in revised manuscript). In the revision, we further quantified intratumoral human T cells by flow cytometry of dissociated tumor tissue using anti-human CD3 antibody and PE staining, confirming PEbody CAR-T cell presence within the tumor microenvironment (**Fig. S6b**).

Functional characterization: the cytotoxic function of PEbody CAR-T cells is presented in **Fig. 1c–f** and **Fig. S1b–d**. We have performed more characterization of PEbody CAR T cells including up-regulation of activation markers CD25 and CD69 (**Fig. S1f**) and secretion of cytokines IFN- γ , TNF- α , and IL-2 (**Fig. S1g**) upon target antigen engagement. Collectively, these additions provide direct validation of in situ AAV delivery and light-induced tCD19 expression and infiltration of PEbody CAR-T cells, and substantiate their functional activity.

Reviewer #3 (Remarks to the Author)

This is a timely article reporting the development of a monobody based PE targeting CAR (PAT) that allows for T cell reactivity to be directed to tumor antigens via PE-conjugated antibodies. The authors demonstrate the ability of PAT T cells to kill tumor cells in a specific manner based on the presence of PE-conjugated antibodies. They further improve the CAR safety profile by engineering the PAT under a light/tamoxifen controlled Cre recombinase (DGLA-PAT CAR), which restricts CAR activity to local illuminated areas and requires the presence of tamoxifen. The authors also engineered a synNotch regulated PAT (sPAT), whereby the PAT expression is induced by priming of the SynNotch antigen. Most notably, the authors devised a CAR-antigen pairing system, whereby tumor antigen expression is controlled by the DGLA system and T cell reactivity is controlled by the sPAT modules (DGLA-sPAT). In this strategy, tumor cells can be vaccinated with an inducible antigen that can then “train” T cells to react specifically at the tumor site through the synNotch module. In murine flank models of human breast cancer, the authors show MDA cells engineered or vaccinated with the DGLA inducible CD19 cassette, express the target antigen only under illumination and with tamoxifen treatment. Administration of sPAT T cells and PE-conjugated PD-L1 antibody resulted in tumor regression only in tumors that were illuminated with light. These efforts are framed in the context of antigen escape, tumor heterogeneity, and safety concerns arising from clinical experience of standard CAR-T therapy, particularly for solid tumors. The impact and novelty of the manuscript is well suited for Nature Communications. The following revision experiments/edits would strengthen the manuscript:

1. In cytotoxicity assays (Figure 1 and Figure 3), the authors should include a conventional CAR control as a benchmark. It is unclear whether the inducible CAR formats reported in this study

are of similar potency as standard CARs used in the field. The authors do a good job of motivating the clinical relevance of the study (i.e. addresses antigen heterogeneity, safety, and the biocompatibility of PE), but having a benchmark for potency is also needed to assess the clinical relevance. In addition to tumor killing, measuring the levels of cytokine secretion (i.e. IFN γ , IL-2) from tumor cocultures is appropriate for assessing T cell functionality.

Response: We really appreciate your suggestion. We agree that benchmarking against a conventional CAR and assessing cytokine secretion are important for evaluating clinical relevance. In the revised manuscript, we have added conventional CAR controls to the cytotoxicity assays (**Fig. 1c,e** and **Fig. 3d,g** in the revised manuscript). We also included ELISA measurements of key secreted cytokines including IFN- γ , TNF- α , and IL-2 after tumor co-cultures (**Fig. S1g**), providing functional readouts complementary to target-cell killing.

To further strengthen this benchmarking, we conducted an in vivo head-to-head comparison between DGLA-sPAT CAR-T cells and conventional CAR-T cells using a PSMA⁺ tumor model (**Fig. S10a**). Both CAR formats achieved comparable tumor control (**Fig. S10e-g**), demonstrating that the DGLA-sPAT design preserves full antitumor potency in vivo. Additionally, in this model, human PSMA was also ectopically expressed in the mouse liver via tail-vein lentiviral transduction (**Fig. S10b-d**) to enable assessment of on-target off-tumor (OTOT) effects. As expected, mice treated with conventional CAR-T cells exhibited pronounced systemic activation, including significant weight loss (**Fig. S10h**), elevated inflammatory cytokines (**Fig. S10i-k**), increased serum alanine aminotransferase (ALT, **Fig. S10l**), a standard indicator of hepatocellular injury, and apparent lymphocytic infiltration with liver tissue damage (**Fig. S10m**). In contrast, DGLA-sPAT CAR-T cells maintained robust tumor-killing activity without inducing OTOT-associated inflammation or tissue injury, underscoring their preserved functionality together with substantially improved safety. Together, the existing in vitro benchmark data combined with the direct in vivo comparison confirm that the DGLA-PAT/DGLA-sPAT inducible CAR-T systems maintain potency comparable to conventional CAR-T cells, while offering a significant safety advantage through spatially restricted activation. We have clarified these points in the revised manuscript (line 274-290).

2. Similar to comment 1, Mock (untransduced) T cells are the appropriate negative controls in cytotoxicity assays that allow the reader to assess the leaky or non-specific activity of the system and should be included.

Response: Mock controls using untransduced T cells have been included in the cytotoxicity assays (**Fig. 1c,e** and **Fig. 3d,g** in the revised manuscript) and noted in the corresponding figure legends.

3. In Figure 3C, are the cells gated on transgene+? It is unclear if the relatively low induction of CD19 CAR-T (~15%) is because of inefficiencies of the inducible system or if it is due to low transduction efficiencies. The authors should clarify these details.

Response: We thank you for the suggestion. For the assays in Fig. 3c, we used FACS-sorted T cells containing the transgenes of the inducible system. Thus, the result reflects transgene positive cells rather than variability from mixed transduction efficiencies. The ~15% induction likely arises from limited expression of the DGLA regulator components in primary T cells, where transgene expression is tightly regulated. This phenomenon is consistent with our previous findings, where we observed that the efficiency of induction is dependent on regulator expression levels in T cells (Huang et al., Sci Adv, 2020). To address this limitation, we subsequently adopted an alternative strategy where the DGLA system is delivered into tumor cells rather than T cells (**Fig. 4a**), which allows for stronger expression of tCD19 antigen and significantly improved induction efficiency (~85%, **Fig. 4e**).

4. The cytotoxicity data presented in this study are normalized. The authors should indicate what the plots were normalized to.

Response: We appreciate your suggestion for clarification. The cytotoxicity data were normalized to tumor-only control wells without T cells. Details regarding this normalization method have been added to the figure legends and Materials and Methods section in the revised manuscript (line 475-476).

5. In Figure 5E, it is unclear if the increase in fluc signal is driven by T cell expansion or synNotch activation since there is no constitutive and orthogonal luciferase that was coexpressed in the T cells. Did the authors administer PE conjugated antibody in this experiment? Adequate

clarification or additional experiments should be provided to help interpret the source of the increasing fluc signal.

Response: We thank you for this insightful question. In the experiment shown in Figure 5E, PE-conjugated antibody was administered one day after T cell injection. We believe that the initial Fluc signal observed is primarily driven by synNotch activation of the injected T cells homing to the tCD19+ tumor site, rather than T cell expansion per se. This is because T cell expansion would require activation signaling, which in our system depends on the synNotch-mediated induction of PEbody CAR followed by its engagement with the target antigen via the corresponding PE-conjugated antibody. It is also important to note that in our synNotch-inducible Fluc design, any newly-generated T cells (resulting from expansion) would still require synNotch activation to express Fluc. Therefore, while T cell expansion may contribute to the overall signal increase over time, T cell expansion alone may not increase the Fluc signal without the synNotch activation. We have clarified this point in the revised manuscript (line 224-228) to ensure accurate interpretation of these results.

6. The authors should add a reference for the following study that demonstrated an inducible CAR regulated by tamoxifen activated recombinase.

Chakravarti, Deboki, et al. "Inducible gene switches with memory in human T cells for cellular immunotherapy." ACS synthetic biology 8.8 (2019): 1744-1754.

Response: We thank you for bringing this relevant reference to our attention. We have added this reference in the revised manuscript (line 294-296) to acknowledge this prior work on inducible CAR regulation.

References

1. Allen, M. E., et al. An AND-Gated Drug and Photoactivatable Cre-loxP System for Spatiotemporal Control in Cell-Based Therapeutics. *ACS Synth Biol* **8**, 2359-2371 (2019).
2. Cabrera, A., et al. The sound of silence: Transgene silencing in mammalian cell engineering. *Cell Syst* **13**, 950-973 (2022).
3. Chappaz, S. and D. Finke The IL-7 signaling pathway regulates lymph node development independent of peripheral lymphocytes. *J Immunol* **184**, 3562-3569 (2010).
4. Di Stasi, A., et al. Inducible apoptosis as a safety switch for adoptive cell therapy. *N Engl J Med* **365**, 1673-1683 (2011).
5. Flugel, C. L., et al. Overcoming on-target, off-tumour toxicity of CAR T cell therapy for solid tumours. *Nat Rev Clin Oncol* **20**, 49-62 (2023).

6. Gendler, S. J. MUC1, the renaissance molecule. *J Mammary Gland Biol Neoplasia* **6**, 339-353 (2001).
7. Holliday, D. L. and V. Speirs Choosing the right cell line for breast cancer research. *Breast Cancer Res* **13**, 215 (2011).
8. Huang, Z., et al. Engineering light-controllable CAR T cells for cancer immunotherapy. *Sci Adv* **6**, eaay9209 (2020).
9. Lee, S. Y., et al. CD8(+) chimeric antigen receptor T cells manufactured in absence of CD4(+) cells exhibit hypofunctional phenotype. *J Immunother Cancer* **11** (2023).
10. Morgan, R. A., et al. Case report of a serious adverse event following the administration of T cells transduced with a chimeric antigen receptor recognizing ERBB2. *Mol Ther* **18**, 843-851 (2010).
11. Morsut, L., et al. Engineering Customized Cell Sensing and Response Behaviors Using Synthetic Notch Receptors. *Cell* **164**, 780-791 (2016).
12. Movahedin, M., et al. Glycosylation of MUC1 influences the binding of a therapeutic antibody by altering the conformational equilibrium of the antigen. *Glycobiology* **27**, 677-687 (2017).
13. Park, A. K., et al. Effective combination immunotherapy using oncolytic viruses to deliver CAR targets to solid tumors. *Sci Transl Med* **12** (2020).
14. Roybal, K. T., et al. Engineering T Cells with Customized Therapeutic Response Programs Using Synthetic Notch Receptors. *Cell* **167**, 419-432 e416 (2016).
15. Slamon, D. J., et al. Human breast cancer: correlation of relapse and survival with amplification of the HER-2/neu oncogene. *Science* **235**, 177-182 (1987).
16. Terakura, S., et al. Generation of CD19-chimeric antigen receptor modified CD8+ T cells derived from virus-specific central memory T cells. *Blood* **119**, 72-82 (2012).
17. Wu, Y., et al. Control of the activity of CAR-T cells within tumours via focused ultrasound. *Nat Biomed Eng* **5**, 1336-1347 (2021).

We thank the Editor and the Reviewers for their careful evaluation of our manuscript and for the constructive and encouraging comments. We have incorporated the requested new data and revised the manuscript accordingly. Below, we provide a point-by-point response to each comment, with changes indicated in the revised version.

Reviewer #1 (Remarks to the Author):

I thank the authors for their comprehensive responses to my previous comments and for the additional experiments provided, which have effectively addressed my concerns.

Specifically, the following:

- 1. Supplementary data on CAR-T cell detection in peripheral blood and spleen, as well as safety experiments including liver and kidney function tests and histopathological studies, strongly support the tumor-local activation characteristics and safety of this system.*
- 2. The addition of isotype controls in the CD19 knockout experiment enhances the reliability of the results.*
- 3. The validation of CD19 and PSMA expression in engineered PC-3 cells is now complete.*
- 4. The explanation of the design change from DGLA-PAT to DGLA-sPAT and related in vivo experiments effectively address questions regarding the control of off-tumor toxicity.*
- 5. Statistical analysis of pre-treatment tumor size and verification that tCD19 expression has no effect on cell behavior eliminate my concerns about potential experimental design.*
- 6. The re-performed CD19-negative cytotoxicity experiment showed no difference between the two groups, consistent with theoretical expectations, making the conclusions more reliable.*

In summary, the revised paper shows improvements in experimental completeness and clarity of argumentation. The method proposed in this study is innovative in improving the specificity and safety of CAR-T therapy.

I believe the authors have responded carefully to the reviewers' comments, and the paper meets the publication requirements; therefore, I recommend acceptance.

Response: We sincerely thank the reviewer for the thorough evaluation and the positive and encouraging comments. We are pleased that the additional experiments and clarifications have adequately addressed the reviewer's concerns.

Reviewer #2 (Remarks to the Author):

All in all, the authors did a great job at addressing my (and it seems like the other reviewers') previous concerns. I commend them for their effort to address my concerns experimentally. This research is now much more robust. Overall, very nice work.

Response: We thank the reviewer for the positive feedback and are pleased that the revisions have addressed the concerns and strengthened the manuscript.

Reviewer #3 (Remarks to the Author):

Please add the ELISA data for conventional CAR in Figure S1G. Otherwise, the revised manuscript has addressed my concerns and I recommend it for publication.

Response: We thank the reviewer for the suggestion and have added the requested ELISA data for conventional CAR in Supplementary Figure S1G of the revised manuscript (line 105).

Response to Reviewers

We thank the Editor and the Reviewers for their careful evaluation of our manuscript and for the constructive and encouraging comments. Below, we provide a point-by-point response to each comment.

Reviewer #3 (Remarks to the Author):

The authors have appropriately addressed my concerns in the revised manuscript. I recommend it for publication and congratulate the reviewers on the excellent work.

Response: We sincerely thank Reviewer #3 for the positive evaluation and encouraging comments. We appreciate the reviewer's recognition of our revisions and are grateful for the support of our work.